



# Ground surface temperatures indicate the presence of permafrost in North Africa (Djebel Toubkal, High Atlas, Morocco)

Gonçalo Vieira[1], Carla Mora[1], Ali Faleh[2]

1) Centre for Geographical Studies, IGOT, Universidade de Lisboa, Portugal.

5  2) Université Sidi Mohammed Ben Abdellah, Fès, Morocco.

*Correspondence to*: Gonçalo Vieira (vieira@campus.ul.pt)

**Abstract.** Relict and present-day periglacial activity have been reported in the literature for the upper reaches of the High Atlas
10  mountains, the highest range in North Africa (Djebel Toubkal – 4,167 m a.s.l.). Lobate features in the Irhzer Ikbi South at
3,800 m a.s.l. have been previously interpreted as an active rock glacier, but no measurements of ground or air temperatures
are known to exist for the area. In order to assess on the possible presence of permafrost, analyse data from June 2015 to June
2016 from two air temperature sites at 2,370 and 3,200 m a.s.l., and from four ground surface temperature (GST) sites at 3,200,
3,815, 3,980 and 4,160 m a.s.l. allowing to characterize conditions along an altitudinal gradient along the Oued Ihghyghaye
valley to the summit of the Djebel Toubkal. GST were collected at 1-hour intervals and the presence of snow cover at the
monitoring sites was validated using Landsat-8 and Sentinel-2 imagery. Two field visits allowed for logger installation and
collection and for assessing the geomorphological features in the area. The results show that snow plays a major role on the
thermal regime of the shallow ground, inducing important spatial variability. The lowest site at 3,210 m showed a regime
characterized by frequent freeze-thaw cycles during the cold season but with a small number of days of snow. When snow
sets, the ground remains isothermal at 0 °C and the thermal regime indicates the absence of permafrost. The highest sites at
3,980 and 4,160 m a.s.l. showed very frequent freeze-thaw cycles and a small influence of the snow cover on GST, reflecting
the lack of snow accumulation due to the their wind-exposed settings in a ridge and in the summit plateau. The site located at
3,815 m in the Irhzer Ikbi South valley showed a stable thermal regime from December to March with GST varying from -4.5
to -6 °C, under a continuous snow cover. The site's location in a concave setting favours snow accumulation and lower
incoming solar radiation due to the effect of a southwards ridge, favouring the maintenance of a thick snow pack. The stable
and low GST are interpreted as a strong indicator of the probable presence of permafrost at this site, an interpretation which is
supported by the presence of lobate and arcuate forms in the talus deposits. These results are still a first approach and
observations through geophysics and boreholes are foreseen. This is the first time that probable permafrost is reported from
temperature observations in the mountains of North Africa.



## 1. Introduction

Permafrost occurrence in arid and semi-arid mountains plays a significant environmental role due to its influence on hydrology. Frozen ground induces refreezing of rain and snowmelt and acts as a subsurface water reservoir able to support streamflow even during the dry season. Its influence can therefore be significant for ecosystems and biodiversity, and in some mountain

areas permafrost may show impacts for agriculture and grazing, but permafrost research in high remote mountain regions is still in its early stages (see Rangecroft et al., 2013). In fast changing and sensitive mountain environments, permafrost niches can also provide special conditions allowing for the occurrence of biological refugia, which may include endemisms and extremophiles of scientific significance (Hu et al., 2015; Jansson and Taş, 2014). Ice-rich permafrost and ground ice may also allow for environmental reconstruction (Lacelle and Vasil'chuk, 2013). Permafrost also plays a significant role for

geomorphological dynamics in mountains, with a number of specific associated landforms and hazards linked to its warming and consequent thaw, such as rock falls and landslides (Haeberli et al., 2010).

The climatic importance of permafrost and the active layer has lead to their classification as Essential Climate Variable (ECV 9) by the Global Climate Observing System of the World Meteorological Organization (Smith and Brown, 2009). The International Permafrost Association maintains the Global Terrestrial Network for Permafrost (GTN-P), which includes

over 1074 boreholes, but with only 31 sites in mountain permafrost settings (Biskaborn et al., 2015), which are still poorly assessed regions (Gruber and Haeberli, 2009).

Contemporary permafrost occurrence is known in the Western Mediterranean region but is mostly constrained to small areas at high altitude or shady sites. In the Pyrenees several active rock glaciers are present and their distribution suggests that the lower limit of permafrost is at about 2,630-2,700 m a.s.l. (Oliva et al., 2016b; Serrano et al., 2009). Geophysical and

temperature observations compiled from several authors by Serrano et al. (2009) suggest that continuous permafrost occurs above 3,000-3,100 m a.s.l. In the Sierra Nevada (37º 03' N, 03º 19' W) an isolated patch of permafrost and a very small rock glacier lobe occur in the Veleta cirque in a north facing cirque at 3,150 m a.s.l., protected by a steep Rockwall and reflecting relict conditions associated with buried ice (Tanarro et al., 2001). The mean annual ground temperature for 1998/99-2008/09 in a shallow borehole in the rock glacier was 0.6 ºC at 0.05 m depth and -1.4 ºC at 1.5 m (Salvador Franch et al., 2011). A deep

borehole in the summit area at the Veleta Peak (3,380 m a.s.l.) shows that permafrost is absent and mean annual ground temperatures were 3.2 ºC at 0.6 m and 2 ºC at 20 m depth.

Periglacial features are widespread in the High Atlas (Hughes et al., 2011) having been described for the Central High Atlas by Couvreur (1966), who has reported active solifluction above 2,200 m a.s.l. The same author states that permafrost is absent in that part of the High Atlas and notes that there is a strong lithological control on the types of periglacial features that occur.

For the Western High Atlas, Chardon and Riser (1981) have pushed the limit of frost activity towards 2,500 m and consider that frost action dominates the morphogenesis above 3,000 m. Robinson and Williams (1992) on a study on sandstone weathering report frequent air temperature minima from -10 to 0 ºC in winter and as low as -20 ºC, at 2,000 m a.s.l. The only landforms supporting a permafrost-related morphogenesis described in the literature are some rock glaciers reported for the





High Atlas by Dresch (1941), Wilche (1953) and Chardon and Riser (1981). Most of them are relict features and at least one case, the Arroumd rock mass near Imlil, has been recently re-interpreted as a very large rock slide (Hughes et al., 2014). The only reference that we have found for active permafrost-related landforms is Chardon and Riser (1981), who interpret lobate features in the Irhzer Ikbi South at 3,800 m a.s.l. as an active rock glacier. However, recent studies are missing and the literature

lacks direct observations and quantitative data supporting the presence of permafrost. The only direct thermal observation of permafrost in the whole Africa is from Mount Kilimanjaro, where permafrost has been reported at 5,785 m a.s.l. with temperatures of -0.03 ºC at 3 m depth (Yoshikawa, 2013).

Given the climate change scenarios that the Mediterranean regions face, marked by warming and precipitation decrease (Giorgi and Lionello, 2007; Montanari, 2013; Simonneaux et al., 2015), permafrost should be close to the threshold of disappearance

in most Mediterranean mountain areas. The subsurface nature of permafrost and the presence of a thawed surface layer in the warmer season (the active layer) strongly limit its identification, characterization and mapping, especially in remote mountain areas (Gruber and Haeberli, 2009).

The present research aims at contributing to solve the question of the presence of permafrost in North Africa and is an exploratory step towards an in-depth assessment aiming at the characterization and modelling of permafrost in the High Atlas.

For such an initial assessment we have installed a set of ground surface temperature (GST) and air temperature data loggers across an altitudinal gradient from 3,200 m to the summit of the Djebel Toubkal in order to characterize the ground temperature regime and heat exchange at the ground-atmosphere interface. The detailed analysis of the GST provides a good insight on the atmosphere-soil interaction, as the major controlling factor on the ground thermal regime.

## 2. Study Area

The Djebel Toubkal is located in the Western High Atlas (31º 4' N, 7º 55' W) and is the highest mountain in North Africa with 4,167 m a.s.l. (Figure 1). The Atlas Mountains comprise a series of ranges and plateaus extending from southwest Morocco to northern Tunisia across more than 2,400 km (Mark and Osmaston, 2008). In Morocco the Atlas Mountains comprise, from north to south: the Middle Atlas (Djebel Bou Naceur, 3,340 m), the High Atlas (Djebel Toubkal, 4,167 m) and the Anti-Atlas (Djebel Sirwa, 3,304 m). The High and Middle Atlas are intracontinental fold-thrust belts located in the foreland of the Rif

(Arboleya et al., 2004). The three major massifs in the High Atlas are, from west to east: the Djebel Toubkal Massif, the Irhil M'Goun Massif (4,071 m) and the Djebel Ayachi (3,751 m).

The climate in the High Atlas is marked by the influence of the Atlantic Ocean to the West, the Mediterranean Sea to the North and the Sahara Desert to the South, resulting in a semi-arid to arid climate (Knippertz et al., 2003; Marchane et al., 2015). The rainy season lasts from November to April and the dry season coincides with the summer, reflecting the Mediterranean style

of the climate (N'da et al., 2016). Annual rainfall exceeds 600 mm above 700 m, with summer precipitation being mostly convective. Boudhar et al. (2014) report an average of 520 mm of annual precipitation for the period of 1989 to 2010 in Oukaimeden at 3,200 m elevation. Snow is present from November to April/May in the highest parts of the mountains, but



with irregular regimes (Badri et al., 1994; Peyron, 1980) and snow is rarely continuous at mid-altitude, with events of snowfall and subsequent melt sometimes happening within one week. However, in the highest reaches, snow cover lasts for several weeks to months (Boudhar et al., 2009). Snowmelt contributes to 15-50% of the stream flow in the Tensift catchment, playing a significant role for irrigation (Boudhar et al., 2009). The low atmospheric humidity and typically subfreezing temperatures

above 3,000 m favour losses by sublimation, which can account up to 44% of snow ablation, while at lower altitudes melting prevails (Schulz and de Jong, 2004). Seemingly, the only perennial snow patch in North Africa, occurs in the northern cliffs of the Tazaghart plateau (3,980 m a.s.l.), close to the Toubkal. This feature is described in various recent papers and was identified by Dresch (1941) together with other features (see Hughes, 2014). Its presence may be related to the high snow feeding area in the plateau above, together with the shelter effect of the steep north-exposed cliff face.

The present study was conducted in the upper reaches of the Oued Ihghyghaye valley, between the marabout of Sidi-Chamharouch and the summit of Djebel Toubkal (Figure 1). The lithology of the study area is composed by Precambrian volcanics, such as Piroxene-bearing doleritic basalts and megaporphyric basalts of the Sidi Chamharouch formation (Zahour et al., 2016) and andesites in the Djebel Toubkal (Cheggour, 2008; Rauh, 1952; Ros et al., 2000). The area shows a typical alpine relief with sharp crests rising above 3,500 m and long deep valleys, with the upper catchments showing evidence of

Late Pleistocene glaciation with frequent landforms such as roches mouttounées and moraines (Chardon and Riser, 1981; Hannah et al., 2016; Hughes et al., 2011; Hughes and Woodward, 2008; Mark and Osmaston, 2008; de Martonne, 1924). Extensive talus slopes and debris cones, together with widespread evidence of frost shattering mark the landscape above 3,000 m.

The detailed study area for ground surface temperatures develops between the Neltner refuge and the Toubkal summit along

the Irhzer Ikhibi-South valley, which is the main climbing route. The valley is a hanging tributary of the Oued Ihghyghayene valley and rises southeastwards of the Neltner refuge above a rock knob with numerous glacier polished outcrops at 3,350-3,400 m. Up valley from the knob, the floor shows a steep longitudinal gradient and is filled by boulderly accumulations, grading in to the distal parts of the talus slopes, which are accumulations of decimetric to metric angular clasts, matrix-supported. These deposits are formed by large boulders in the south slope and formed by smaller boulders in the north slope.

The deposits show lobate forms and incipient ridges and furrows at ci. 3,800 m suggesting active periglacial dynamics. Chardon & Riser (1998) interpret this sector as an active rock glacier. Slopes surrounding the Irhzer Ikhibi South valley are steep, with angular taluses with free faces in the rock knob area, along the south slope and in the Toubkal face. Toubkal west col and most of the slope north of valley are debris mantled. The south ridge rises above 3,900 m causing a significant shadowing effect during the winter in the valley floor. A snow patch occurs frequently until June in the Irhzer Ikhibi South valley, especially in

the Toubkal slope and may be partly responsible for the debris ramparts, first described by Chardon & Riser (1998) that occur at 3,800 m. Google Earth imagery allows identifying around the Toubkal numerous debris-mantled slopes and taluses with flow-like lineaments, suggesting creep, and small rock glacier-like features are also identifiable. At the col of the Irhzer Ikhibi Nord just north of the Toubkal, at 3,900 m, solifluction lobes are present.



## 3. Methods

### 3.1 Air and ground surface temperature monitoring

Air temperature, relative humidity and ground surface temperature data loggers were installed in June 2015 from Sidi Chamharouch (2, 370 m) to Djebel Toubkal (4,160 m a.s.l.) across an altitudinal transect aiming at an hourly characterization

of the soil and air climate for 2015-16. For air temperature and relative humidity, we used Hobo ProV2 loggers, with an accuracy of ±0.2 °C, installed in radiation shields at ci. 2 m height. One was installed close to a shop in Sidi Chamharouch (2,370 m) and the other near the Neltner refuge of the Club Alpin Français de Casablanca (3,210 m). Both sites were surveyed by local partners. A minilogger ibutton DS-1922L was installed close to the summit of Djebel Toubkal (4,160 m), hidden in a shaddy location in the iron trig that stands at the top. However, this logger disappeared and data was lost.

For GST, single channel miniloggers Hobo TidBit with an accuracy of ±0.2 °C were glued to the lower face of a 15x15x0.2 cm high diffusivity steel plate that maximizes contact with the soil particles, when buried at 2-3 cm depth (see Ferreira et al. 2016). In order to check for drifts on temperature accuracy after retrieval, the loggers were tested under various temperature settings (-20 °C to 39 °C) and showed average differences under 0.1 °C, which is well within sensor error. Four of such plates were used between the Neltner refuge and the summit of Djebel Toubkal. The sites were chosen in order to characterize the

altitude control on GST and were installed along the main climbing route, with care in order to avoid stepping and surfacing due to the large number of climbers. All sites were installed in stony silty-sandy soils, matrix-supported, in gentle slope positions, but where locally the terrain was relatively flat. Neltner was installed above the refuge in a boulderly diamicton. Toubkal 3 was installed in boulderly diamicton in a valley position, where snow accumulation is favoured. Toubkal 2 was installed in a debris-covered slope, a few meters below a ridge crest. Toubkal 1 was installed in the debris-covered surface of

the Djbel Toubkal summit plateau, about 100 m from the summit, in order to avoid the proximity to climbers. Details of the sites are provided in Table 1 and figure 2.

Temperatures were recorded hourly from 16 June 2015 to 16 July 2016 with the objective of having a whole year of data, centered in the cold season. All measures presented are derived from hourly data. No other loggers were installed due to funding limitations and high probability of disappearence at high altitude sites.

### 3.2 Remote sensing characterization of the snow cover

Although ground surface temperature regimes allow for identifying the presence of snow cover with high degree of confidence, for this study it is essential to demonstrate that snow played a major role on GST and that it was present at some of the sites. For characterizing the snow cover we used 18 scenes from Landsat 8 OLI and 2 scenes from Sentinel 2-A collected between 16/09/2015 and 14/06/2016. Landsat scenes were obtained at 16-day intervals, at 10 AM local time (USGS, 2016)and only

30 one scene showed partial cloud cover. Sentinel-2 scenes at 10:30 AM complement the series and allow confirming the results obtained with Landsat. Pixel size is 30 m for Landsat and 10 m for Sentinel 2 (ESA, 2015). We have used full resolution georeferenced visible colour composites provided by USGS at EarthExplorer. The images allowed to identify the general snow





cover conditions at the monitoring sites. For the interpretation, caution was necessary since the spatial resolution of the imagery is much larger than the microscale variability that may affect the monitoring sites. The snow conditions at each site were classified by visual inspection of the imagery as « no snow », « possible snow/snow margin », « snow » and « significant snow ». The classification was done on-screen in QGIS using an overlay of elevation contours and the coordinates of the

monitoring sites for better accuracy. Differences between snow and cloud cover were easily identifiable and clouds were rare.

### 3.3 Climate series and extrapolation

In order to assess the climatic representativity of the period of June 2015 to July 2016, a long-term climate series of temperature and precipitation from a nearby meteorological station is needed. This allows comparing monthly records with the reference series and better frame the study period and discussing the results. However, the High Atlas has no long-term meteorological

stations and the regional network is very sparse. The only long-term meteorological data available are from Marrakesh (Menara) in the plains north of the mountain range at 468 m a.s.l. and from Ouarzazate, in the southern piedmont, at 1,153 m a.s.l, but in a very dry setting. Middelt, located 350 km to the east of the Toubkal Massif at 1,515 m a.s.l. shows very incomplete data. Given the more complete data series of Menara, and strong correlation of the mean monthly air temperatures with those that we have measured at Sidi Chamharouch and Neltner ($r^2 = 0.94$, p < 0.000) (Figure 3), we use it for the analysis. The data

was obtained from the Custom Monthly Summaries of the Global Historical Climate Network (NCDC) for 1977-2016. Before 1977 there are several gaps in the series.

## 4. Results

### 4.1 Climate characteristics of the study period

Climate records from Marrakesh (Menara – Figure 4) for June 2015 to March 2016 (no data available afterwards) show that

mean monthly temperature from June to December was close to the median, but January was extremely warm, with a value (15.2 ℃) close to the maximum (15.5 ℃) of the period 1977-2015 and well above the 3rd quartile (13.4 ℃). February with 14.9 ℃ was between the median and the 3rd quartile and March showed a mean monthly temperature close to the 1st quartile, with 15.4 ℃. Precipitation showed very high values in August and September (close to the maximum), decreasing afterwards, with November to January as very dry months, below the 1st quartile and close to the minimum. February showed precipitation

close to the median and March, close to the 3rd quartile. The study period was therefore initially characterized by a wet summer, followed by a dry autumn and early winter, which coincided with a very warm January, followed by a very cold and wet March.

The mean monthly air temperatures measured at Sidi Chamharouch and Neltner show a good correlation with Menara, with the exception of February and March, which were colder in the mountain massif (figure 5). The very high correlation between

both air temperature sites ($r^2 = 0.99$, p<0.00) allowed to estimate an average temperature lapse rate of -0.59 ℃.100 m$^{-1}$ and therefore by linear extrapolation, to estimate the temperatures at the summit of Djebel Toubkal for the study period. These



results are close to the -0.56 ℃.100 m⁻¹ calculated by Boudhar et al. (2009) using a weather station data from Saada (411 m) and Oukaimeden (2,760 m) for 1998-2005. The mean annual air temperature at the Toubkal was 0.6 ℃, with the minimum mean monthly temperature occurring in February with an estimated -6.8 ℃. November to April show estimated mean monthly temperatures below 0 ℃.

Reports from local guides indicate that the winter of 2015/16 was anomalous for snow conditions, with a very late onset of the snow pack in mid-February. This is confirmed by the remote sensing data (figure 6), which shows snow in early October 2015 affecting the Toubkal massif with a peak in the scene of 18/10/2015, then decreasing progressively until early January when a short duration cover shows up, then melting again until 7 February. It was only between 7 and 23 February that significant snowfall occurred covering the whole study area. The snow cover remained in the valley floors and concave areas until 11

April, but quickly melted from the ridges and south facing slopes. In mid-May another large snow fall event took place, but snow melted quickly in two weeks and by mid-June was completely gone from the study area.

## 4.2 Ground Surface Temperatures

Mean monthly ground surface temperatures at the Toubkal massif showed a similar annual regime at the 4 sites (figure 7),

with the warmest month in June 2016 showing values from 13.5 ℃ at T2 to 17.7 ℃ at NLT. The cold season showed mean monthly temperatures below 0 ℃ from November to March at the 3 higher sites, with NLT still showing a positive average in November. T3 showed stable mean GST below -5 ℃ from December to March, while T1 showed the minimum in February with -4.5 ℃.

The daily absolute maxima and minima follow in general the regime of the monthly means, except at T3 and evidence very

high amplitudes. A plateau in the maxima is observable from April to September, with T3 showing the highest values reaching 39.3 and 40.5 ℃ in June and July 2016, respectively. At the same site, the stabilization of the absolute maxima below -4.4 ℃ in January and February 2016, as well as the sudden rise from 0 ℃ in March to 31.3 ℃ in April, are noteworthy.

The mean annual GST (MAGST) shows a statistical significant linear correlation with altitude ($r^2 = 0.94$, p<0.00), with a rate of ci. -0.4 ℃.100 m⁻¹, with T3 showing a residual of -0.4 ℃ and with the other sites aligned with the straight line (figure 9).

MAGST was 6.2 ℃ at NLT, 3.2 ℃ at T3, 3.4 ℃ at T2 and 2.8 ℃ at T1, the summit of Toubkal.

Four altitudinal patterns of mean monthly GST occurred in the study period (figure 9). Group 1 includes the warm season, with June, July and August and shows decreasing GST with altitude, with T2 slightly colder than T1. Group 2 integrates the transition months, with May, September and October showing a similar continuous decrease of GST with altitude and a rate of ci. -0.47 ℃.100 m⁻¹. Group 3 includes the cold season, with January, February, March, November and December and shows

a general decreasing of GST with altitude, but with T3 being the coldest site. Finally, April 2016 shows up as an outlier, with an inverted rate with altitude, but with T3 as the warmest site (Group 4).

The hourly records of GST allow for a more accurate analysis of the conditions influencing the monthly means and to assess the environmental controls on the ground thermal regime (figure 10). The first striking characteristic of the GST at the 4 sites



is the large diurnal thermal amplitude range, especially from May to September with averages of 11.4 to 12 ℃, except at T3, with 19.7 ℃. Maximum amplitudes were from 19.2 to 22.5 ℃, except at T3, with 35 ℃.

The cold season is clearly defined in the GST, lasting from mid-October to the end of April, with May being a transition month. Differences in GST regimes are clear during the cold season. NLT showed a long period from mid-October to mid-February with small thermal amplitude and numerous freeze-thaw cycles. Afterwards, temperatures remained stable just below freezing until mid-April. At T3 the cold season was marked by subfreezing temperatures from mid-October to mid-April, with temperatures decreasing regularly until mid-December after a zero-curtain effect lasting about 2 weeks in late-October. Then, a stable value starting at around -4.5 ℃ and decreasing irregularly to about -6 ℃ occurred, as situation which lasted until late March, when temperatures increased quickly and then stabilized for about 10 days with a zero-curtain effect. During this period the diurnal amplitudes of GST at T3 were typically between 0.4 and 1.0 ℃, with an average of 0.8 ℃. The GST regime is especially significant in this paper and will be analysed in more detail below. The two upper sites, T2 and T1, show very similar GST regimes during the cold season, with the main differences being the number of freeze-thaw cycles which is larger at T2, with lower maxima during the cold season at T1. Both sites show a short zero-curtain effect in late-October, simultaneously to T1. GST daily amplitudes from mid-December to mid-March averaged 5.2 ℃ at T2 and 4.8 ℃ at T1 and were variable at NLT.

After an increase in GST in April 2016, May showed a significant cooling that lasted for 8 days in NLT and 19 days at T3. This cold event resulted in a stabilization of GST at 0.5 ℃ at NLT, at 0.2 ℃ at T3 and at 0 ℃ at T1 and in a cold but unstable regime at T2.

## 5. Discussion

The hourly GST data across the altitudinal gradient in the Toubkal Massif allows for a detailed insight into the ground thermal regime of the High Atlas periglacial zone. The comparison of the snow cover at different altitudes derived from the remote sensing imagery with the GST regimes confirms that the presence of the snow pack is the cause for stable temperatures close or below 0 ℃ during the cold season at the different sites (figure 6). This fact is clear both for the period from late October to mid-April, but also for the short cooling event in May. The effect is well known and has been shown for different regions. While a thin and compacted snow layer a few centimeters thick allows for fast heat transfer between the ground and the atmosphere, and in some cases, even to an increase in ground cooling due to high albedo and high ice thermal diffusivity, a thick snow pack acts as buffer between the ground and the atmosphere. If snow is thick enough, the thermal wave will be delayed in the ground (Goodrich, 1982; Williams and Smith, 1989). If the ground is unfrozen at depth, heat will flow towards the snow pack and the ground surface temperature will be controlled by phase change at the snow interface, generating near 0 ℃ isothermal regimes (Vieira et al., 2003) - the so-called zero-curtain effect (Outcalt et al., 1990). On the other hand, if the ground is colder than the snow pack, it will generate a heat sink at depth, inducing a decrease in GST, which are function of



the insulating capacity of the snow pack, that also depends on forcing induced by the atmosphere at the snow-air interface (see Haeberli and Patzelt, 1982; Ishikawa, 2002).

Site T3 shows a remarkable regime with relatively stable GST at ci. -5.8 ℃ from December 2015 to March 2016. The small temperature range reflects the insulating effect of snow cover, but a small diurnal heat transfer effect is identifiable in the GST
with daily amplitudes of up to 1 ℃. As an example, the detailed examination of the hourly temperatures at T3 from 4 to 20 December 2015 shows GST consistently below estimated air temperatures, except for a few hours in 11 and 12 December (figure 11). This situation occurred all over the snow cover period. It is also noticeable that the thermal wave is delayed in the snowpack for about 5-6 hours with the resulting GST curve being much smoother than air temperatures.

Ishikawa (2003) reports two sites with GST regimes very close to the one of T3 in the Hidaka Mountains (Hokkaido, Japan),
located in low altitude openwork boulder deposits, favouring cold air drainage and funneling in winter. The author has classified this type of settings as the extrazonal permafrost zone. Lambiel and Pieracci (2008) also report a very similar thermal regime to T3 at the base of a talus slope at low altitude in the Western Swiss Alps, reporting GST in the end of the winter below -5 ℃ as an indicator of the probable presence of permafrost. Values of spring GST below thick snow covers under -3 ℃ are frequently used in the literature as an indication of the probable presence of permafrost, an approach derived from the
widely used Bottom Temperature of Snow (BTS) method (Hoelzle, 1992; Lewkowicz and Ednie, 2004). Numerous cases with sporadic permafrost developing in talus slopes have been reported and the cooling process below coarse debris is well-known (e.g. Delaloye et al., 2003; Delaloye and Lambiel, 2005; Gadek, 2012; Sawada et al., 2003). Gądek and Kędzia (2008) in a study for the Tatra Mountains (Poland and Slovakia), where different GST regimes were analysed, associated sites with steadily decreasing temperatures during the winter to increasing snow cover, and where GST dropped to below about -5 ℃, permafrost
was found. Such sites were not open work, which limited cold air flow through the boulders. Furthermore, small variations in GST through the cold season indicated that the snow insulation was not perfect. In those situations, ground cooling was attributed to the concave locations with cold air flowing over the surface, which cooled the snow and ground. This process acted together with the low summer solar irradiation to promote cooling, with a larger influence on low GST than snow thickness or elevation. Other authors have indicated snow redistribution by avalanches as an important factor promoting cooler
ground and permafrost formation at the base of slopes, since snow lasts longer at those localities, mitigating ground surface warming (Haeberli, 1975). Onaca et al. (2015) indicate this as an important effect for the maintenance of permafrost in the Retezat Mountains (Romania), where permafrost occurs at sites with MAGST below 0 ℃, BTS lower than -3 ℃ and ground freezing index higher than 600 degree-days.

Following the literature, the low and stable GST measured at T3 (3,815 m) are a strong indicator of probable permafrost
occurrence. At the site, the surface material is a boulderly matrix-supported deposit, with a pebbly-sandy matrix and no visible voids. This limits the interpretation of a talus-ventilation related origin for the low GST and suggests conditions close to those described by Gądek and Kędzia (2008) in the Tatra Mountains and also by Onaca et al. (2015) to the Tatras. It is also possible that the deposit is open work at depth, which could help explain the low temperatures, but this hypothesis lacks verification.





The GST regimes measured at Neltner (3,200 m) show clearly that permafrost is absent at the site. The snow pack settles only in mid-February and temperatures become very stable close to 0 ℃, which shows that there is no heat sink at depth, but rather a warmer unfrozen ground. Snow lasted about a week longer than at T3, possibly due to local effects, such as shadowing or thicker snow accumulation associated with snow drift.

The upper sites at the ridge (T2 – 3,980 m) and summit of Toubkal (T3 – 4,160 m) show small events of stable GST as a reaction to the scarce number of snow fall episodes, but long lasting zero-curtains are not visible after the initial one in late October, which was synchronous at T1, T2 and T3 (figure 12). After that episode, GST shows irregular curves with ranges of ~1 to 8 ℃ reflecting an absent or very thin and compact snow layer with high conductivity. The lack of zero-curtain effects and the frequent freeze-thaw cycles at T2 indicate the absence of snow and also a very dry soil. At both sites, the data suggest

that the ground remains frozen below the surface during the cold season, however the GST regime analysis does not allow assessing on the presence of permafrost.

The very high daily GST ranges during the warm season are explained by the high insolation of the Mediterranean high mountain, together with the scarce moisture and rocky nature of the soil. We have no data to explain with safety the very high GST at T3, but they should be function of local differences in soil thermophysical properties, together with the concave setting of the site, receiving more reflected and emitted radiation from the surrounding slopes and also due to a wind shelter effect.

Both T2 and T1 are convex and very wind exposed sites, which may explain lower maxima than at T3.

The probable presence of permafrost at site T3 is not unexpected, since the valley shows landforms typical of mountain permafrost, such as lobate deposits and poorly developed transverse ridges and furrows in the lower part of the scree slope (figure 12), a fact that has been also described by Chardon and Riser (1981). The high altitude and sheltered location favouring

longer lasting snow cover, will also contribute to increased water availability in the soil surface, promoting refreezing within the soil, with the possibility for the presence of interstitial ice. This would facilitate the creep of the frozen talus and the formation of solifluction and creep features that are detectable.

Given the high and statistical significant correlation found between Menara, Sidi Chamharouch and Neltner and the lack of data on climate history at the High Atlas, we have applied the average lapse rate to the long-terms records from Menara and

extrapolated them to the summit of Djebel Toubkal. A similar method has been applied by Hannah et al. (2016) for paleoequilibrium line altitude estimation in the region. The results should be interpreted with care, but it is worth noting that the extrapolation of the warming trend observed in Marrakesh, represents a gradual shift towards positive MAAT at the summit (Figure 13). This suggests that the probable permafrost sites in the High Atlas are possibly at risk of disappearing if the trend is to be continued.

**6. Conclusions**

The analysis of hourly ground surface temperatures from June 2015 to July 2016 across an altitudinal transect in the Toubkal Massif provided, for the first time, data for the high altitude periglacial domain of the High Atlas mountains. The period was



marked by high temperatures in December and January, with very low precipitation from November to January, which caused a late onset of the winter snow cover when compared to other years. GST from 3,210 to 4,160 m a.s.l. showed two very contrasted periods: a hot season from late-May to late-September, and a long cold season, from mid-October to mid-April. The hot season showed positive air temperatures at all sites and was marked by very high daily temperature amplitudes, with

maxima reaching 40.1 ºC at Toubkal 3. This regime was controlled by the high solar radiation inputs and by the dryness of the soil. The cold season was marked by subfreezing GST or by frequent freeze-thaw cycles, depending on snow conditions. Neltner in a valley floor at 3,210 m showed frequent freeze-thaw cycles during the start of the cold season, until a heavy snow fall event in mid-February, inducing GST to stabilize close to 0 ºC. The high altitude sites T2 (3,964) and T1 (4,160 m) located in wind swept areas showed subfreezing temperatures with frequent freeze-thaw cycles illustrating the lack of an insulating

snow cover. These conditions prevailed during the whole cold season, with mean monthly GST below 0 ºC from November to March. The monitoring site Toubkal 3, located in a valley in the lower section of a talus slope at 3,820 m a.s.l. showed the most remarkable GST regime during the cold season. After an early onset of a stable snow pack in mid-October, GST decreased regularly until mid-December and then showed minor oscillations around ci. -5.8 ºC. These conditions were only interrupted in late-March with snow melt, and were followed by a zero curtain effect in April.

The low GST at Toubkal 3 with values around -5 ºC with low thermal amplitude, under the snow pack, show the likely presence of permafrost. The site is located in a valley with long lasting snow cover, facing west, but close to a slope that suffers the shadowing effect of the ridge located southwards. The boulderly surface of the lower talus shows poorly developed ridges and furrows, which are in agreement with the presence of permafrost.

The results presented here have to be interpreted with care and although the data suggests the presence of permafrost, for a

more accurate assessment, more observations are needed. For the continuation of this research, forthcoming studies should target at: i. Electrical resistivity surveying in order to identify possible anomalies and the presence of ice, ii. Installing a larger number of GST loggers at different settings around T3, iii) Installing an air temperature logger at T3 in order to assess the possible shadowing effect on air temperature and GST cooling, and iv) installation of a borehole for monitoring temperatures. The possible presence of permafrost at 3,800 m in the Djebel Toubkal massif, to be confirmed, could become spatially

significant, since such altitudes are frequent in the High Atlas, not only in the Toubkal region, but also further east, in the M'Goun massif. The presence of permafrost would also need to be assessed for: ice-content and consequent impacts for hydrology (at least at a local level), thermal state and spatial distribution at the regional scale, ecological significance (vegetation communities, endemisms, refugia), possible presence of extremophiles and possible analysis of permafrost ice as a paleoenvironmental archive. Given the warming trend as shown by the Menara data-series, future climate scenarios and the

very sensitive setting of the upper reaches of the High Atlas, just above 0 ºC MAAT, further research at an interdisciplinary level is needed, since the possible permafrost remnants could quickly disappear facing climate change. And this can be the last permafrost remnant in North Africa.





## Acknowledgements

Special thanks are due to Philip Hughes, who provided valuable advice on routes and logistics for the Toubkal massif and information on the geomorphology of the region. Warmest thanks to Sebastião Vieira, who participated in the field season of 2015 and helped with the setting of the loggers, for his fantastic company and patience. Mustapha Asquarray (Dar Assarou)

from Imlil provided excellent guiding services and support in the Toubkal region. Ibrahim ait Tadrart Buff and Mohammed Boyikd provided transport support. To the Ait Elkadi family guards of the Neltner Hut and to Hsin Nsliman at Sidi Chamharouch Cafe, we thank for the surveillance of the air temperature/RH loggers. Hassan Michalou is thanked for helping with the air temperature minilogger at the summit of Toubkal. This research benefited from support of the bilateral project COLDATLAS – « Does permafrost occurs in the high mountains of North Africa? » funded by the FCT/CNRST (Portugal-

Morocco). Partial support was provided by the CEG/IGOT-Universidade de Lisboa.

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

| Sites | Alt (m) | Model | Variables | Interval (h) |
|---|---|---|---|---|
| Sidi Chamharouch | 2,370 | Hobo Prov2 | AT, HR | 1 |
| Neltner | 3,200 | Hobo Prov2, TidBit | AT, HR, GST | 1 |
| Toubkal 3 | 3,815 | TidBit | GST | 1 |
| Toubkal 2 | 3,980 | TidBit | GST | 1 |
| Toubkal 1 | 4,160 | Ibutton, TidBit | AT, GST | 4,1 |

**Table 1 – Temperature data loggers installed in the High Atlas. Variables: GST -Ground surface temperature, AT - Air temperature, HR - Relative humidity.**





**Figure 1 – Location and topography of the Toubkal Massif study area. Yellow circles are the sites of the data loggers. Contour equidistance is 50 m.**



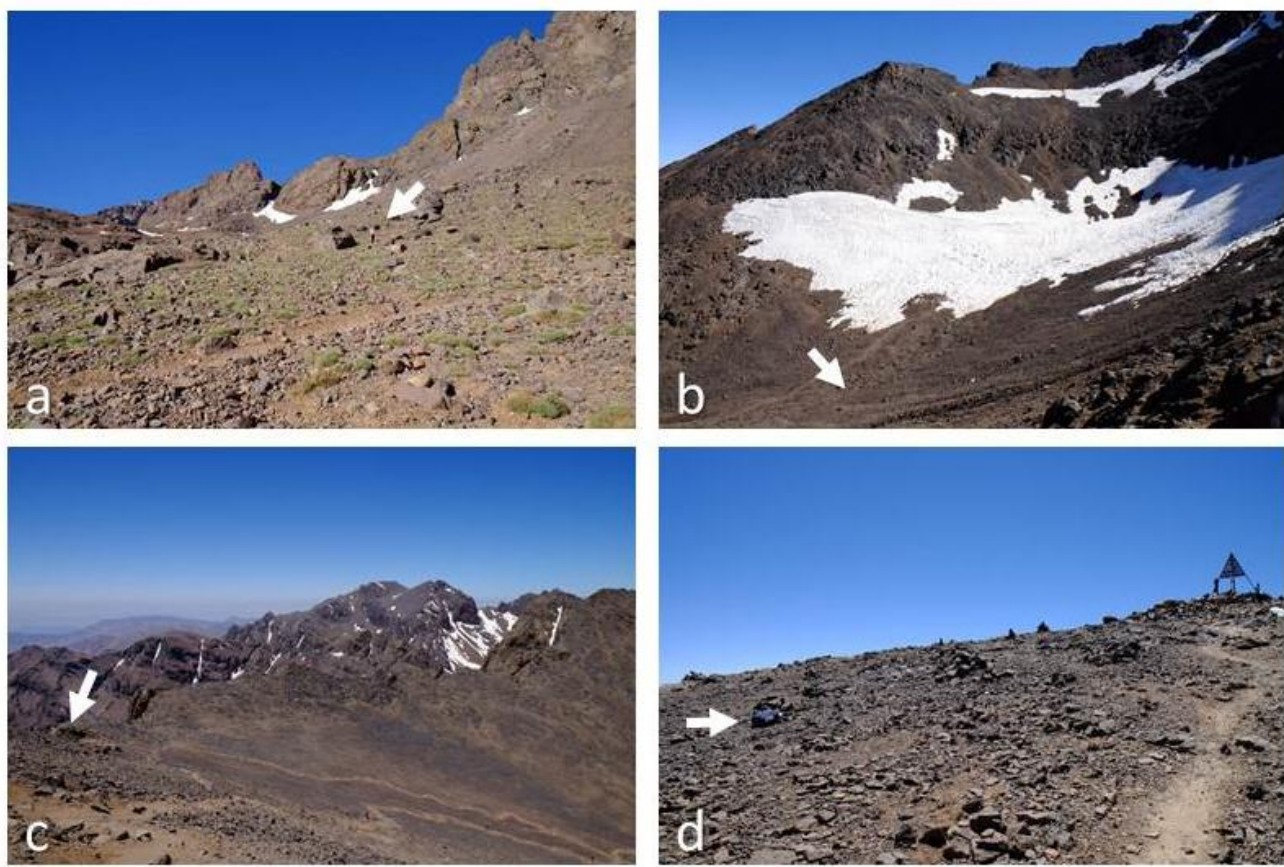

**Figure 2: Location of the ground surface temperature loggers: a. Neltner (3,200 m), b. Toubkal 3 (3,815 m), c. Toubkal 2 (3,980 m), d. Toubkal 1 (4,160 m).**





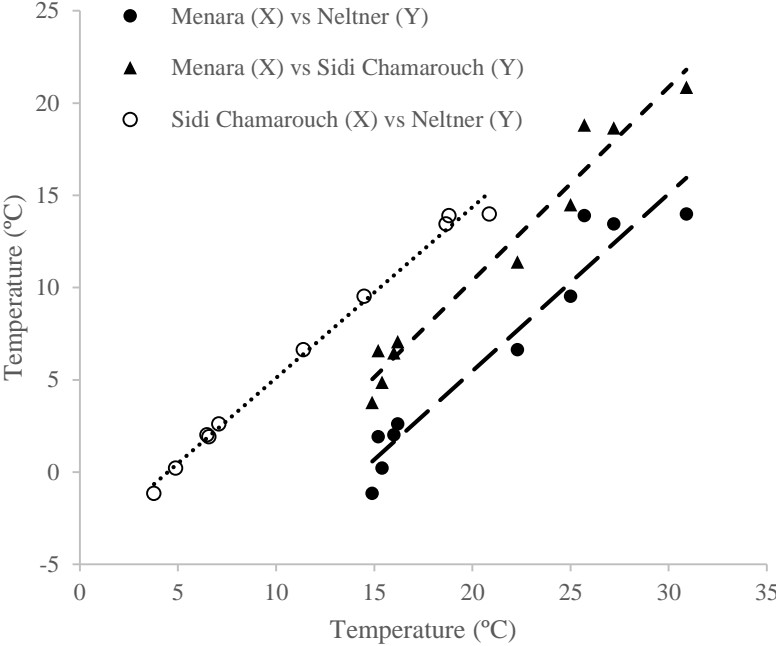

**Figure 3: Correlation between mean monthly air temperatures in Menara, Sidi Chamharouch and Neltner (Data from July 2015 to June 2016 in Sidi Chamharouch and Neltner, July 2015 to March 2016 in Menara).**





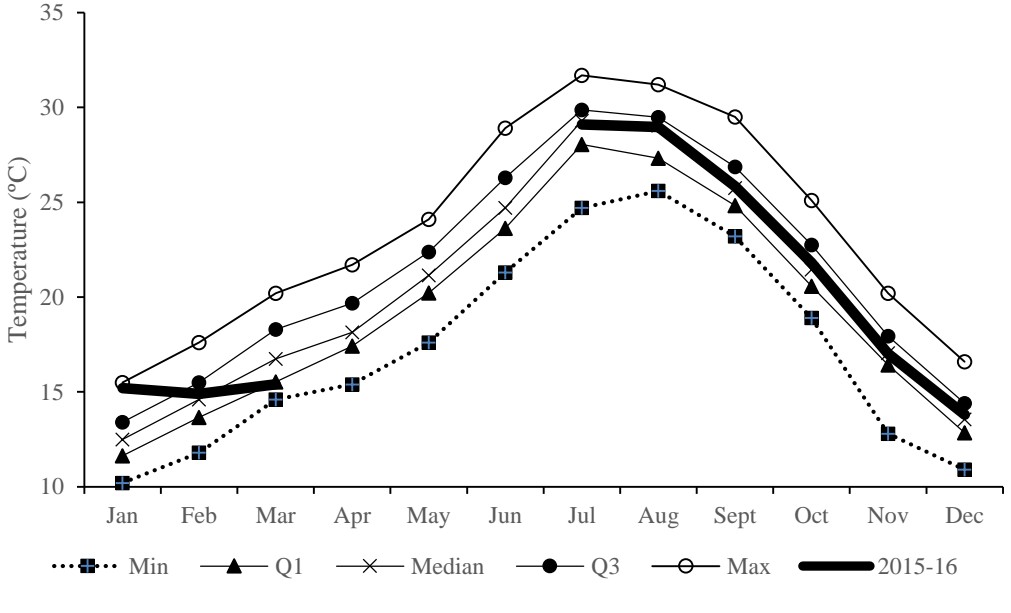

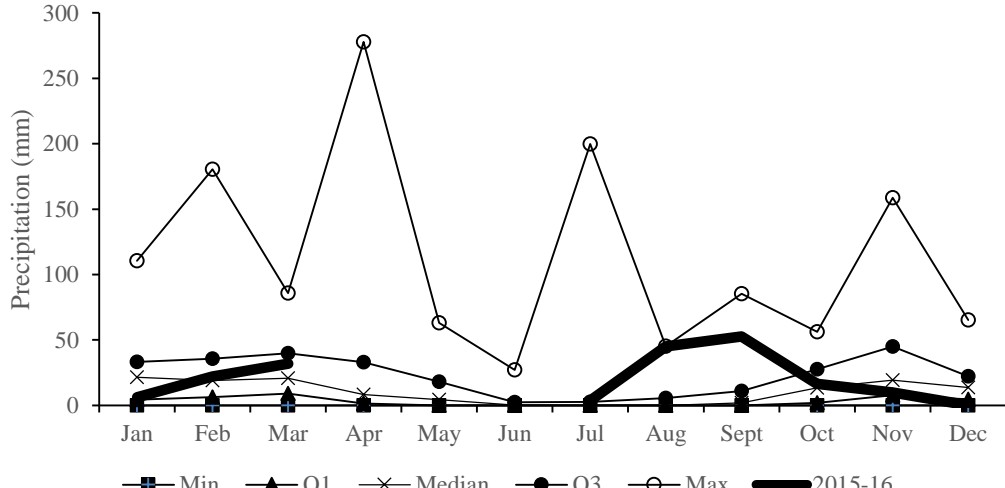

**Figure 4: Statistics of monthly temperature and precipitation at Menara (Marrakesh) from 1977 to 2015 and records of the study period (July 2015 – March 2016). Source: NCDC/GHCN.**





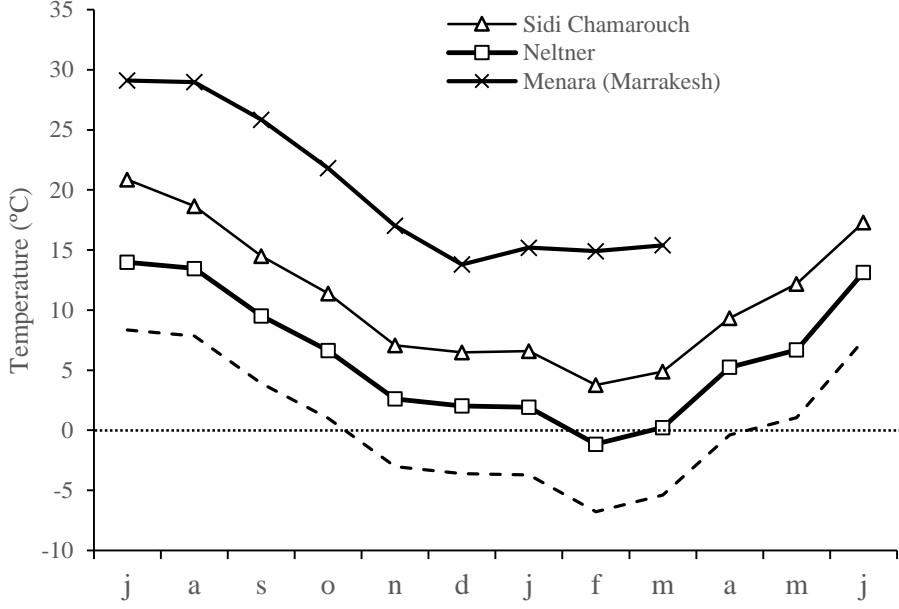

**Figure 5: Mean monthly air temperatures from July 2015 to June 2016 in Marrakesh (Menara), in the two study sites and extrapolated to the summit of Djebel Toubkal (4,167 m a.s.l.).**





**Figure 6: Satellite scenes from Landsat-8 (USGS) and Sentinel-2 (ESA) used for assessing the snow cover at the monitoring sites from September 2015 to June 2016. Satellite imagery obtained from USGS EarthExplorer.**





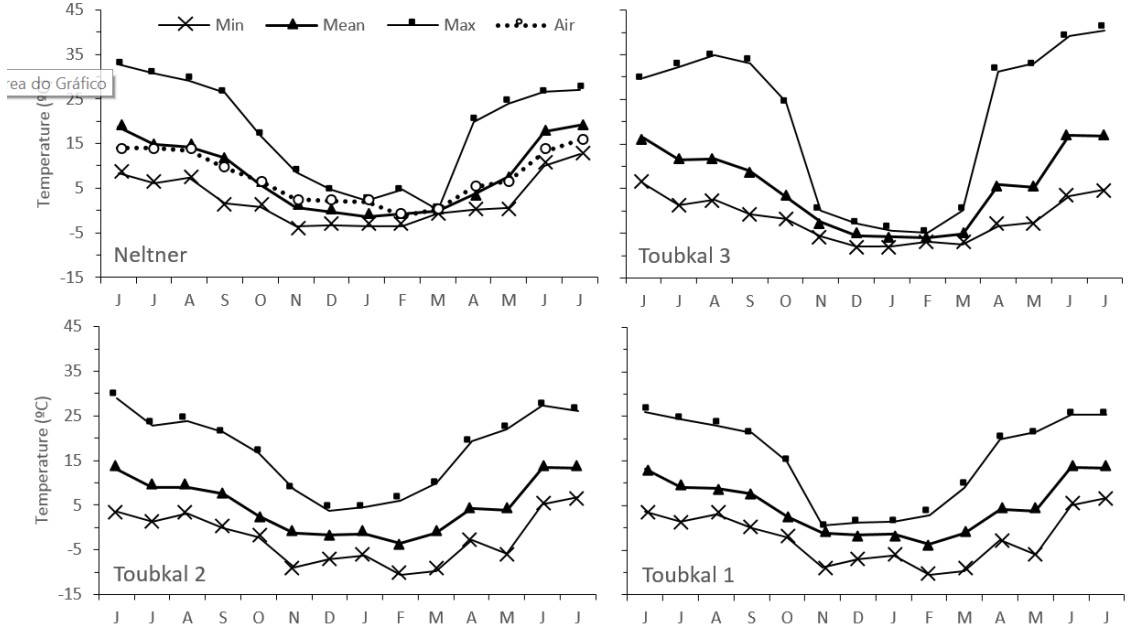

**Figure 7: Monthly ground surface and air temperatures in the Djebel Toubkal massif from June 2015 to July 2016. Extremes are daily hourly maxima and minima.**

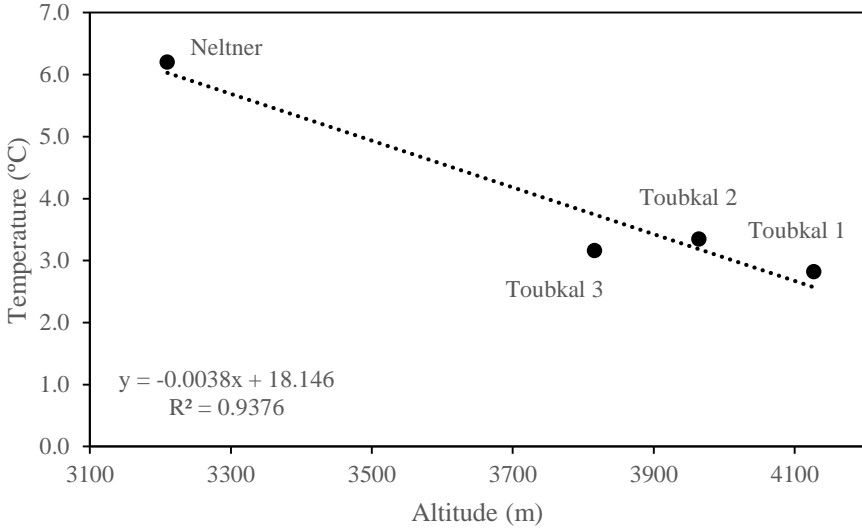

**Figure 8: Mean annual ground surface temperatures vs altitude for the 4 monitored sites in the Djebel Toubkal massif.**



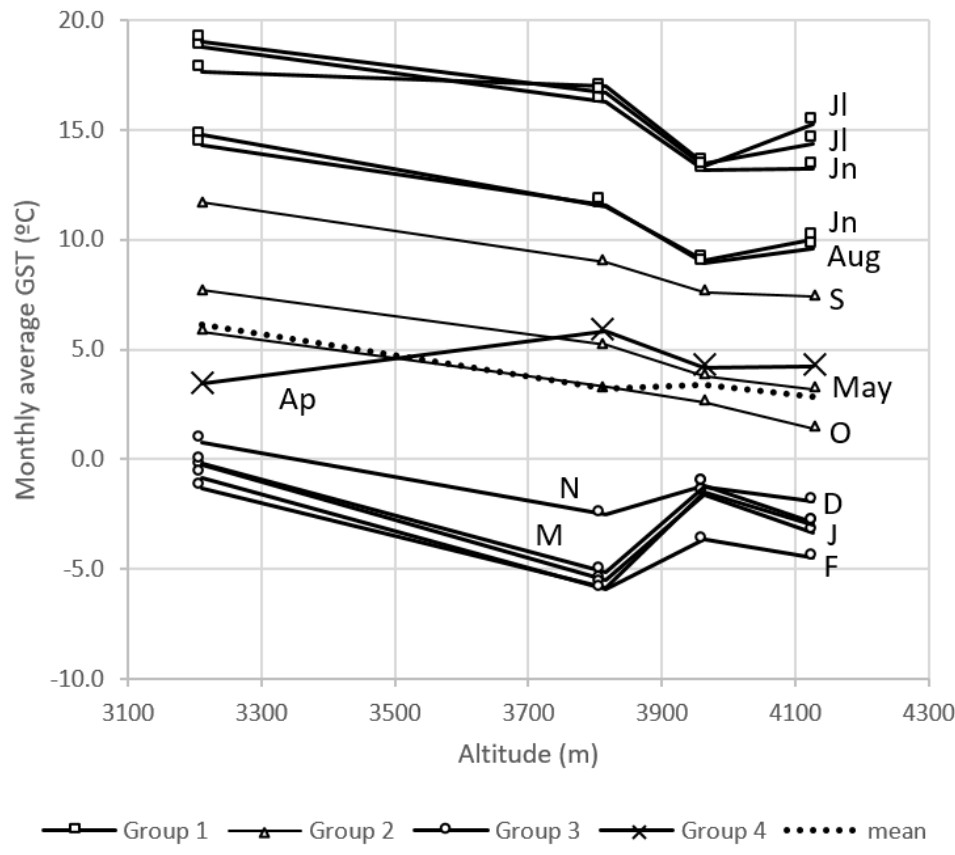

**Figure 9: Mean monthly ground surface temperatures at the 4 study sites in the Toubkal Massif from July 2015 to June 2016.**





**Figure 10: Ground surface temperature regimes at the 4 sites in the Toubkal Massif and air temperature at Neltner. The plot shows hourly data. Symbols indicate the snow cover conditions at the date of satellite scenes: - no snow, + possible snow/snow margin, ++ - snow, +++ - significant snow, ? – Uncertainty in classification.**



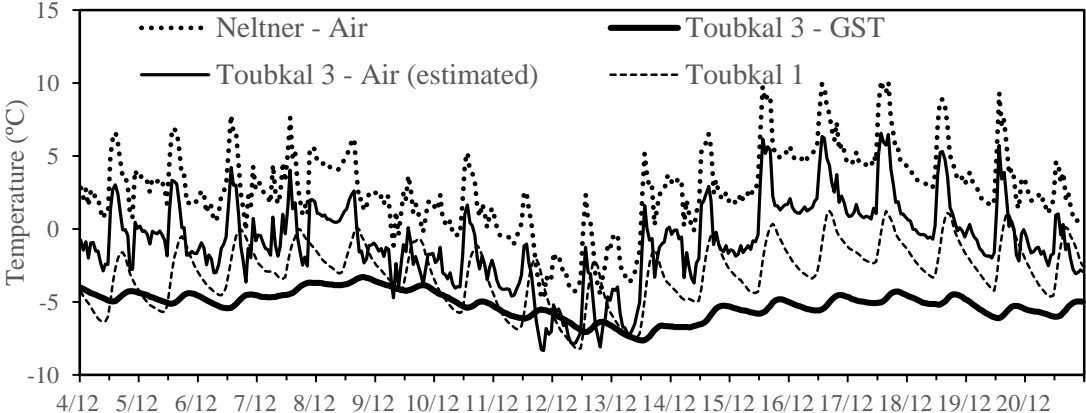

**Figure 11: Example of the hourly GST regime at Toubkal 3 and comparison with air temperatures measured at Neltner and extrapolated to T3 using an observed lapse rate of -0.59 ºC.100 m$^{-1}$.**

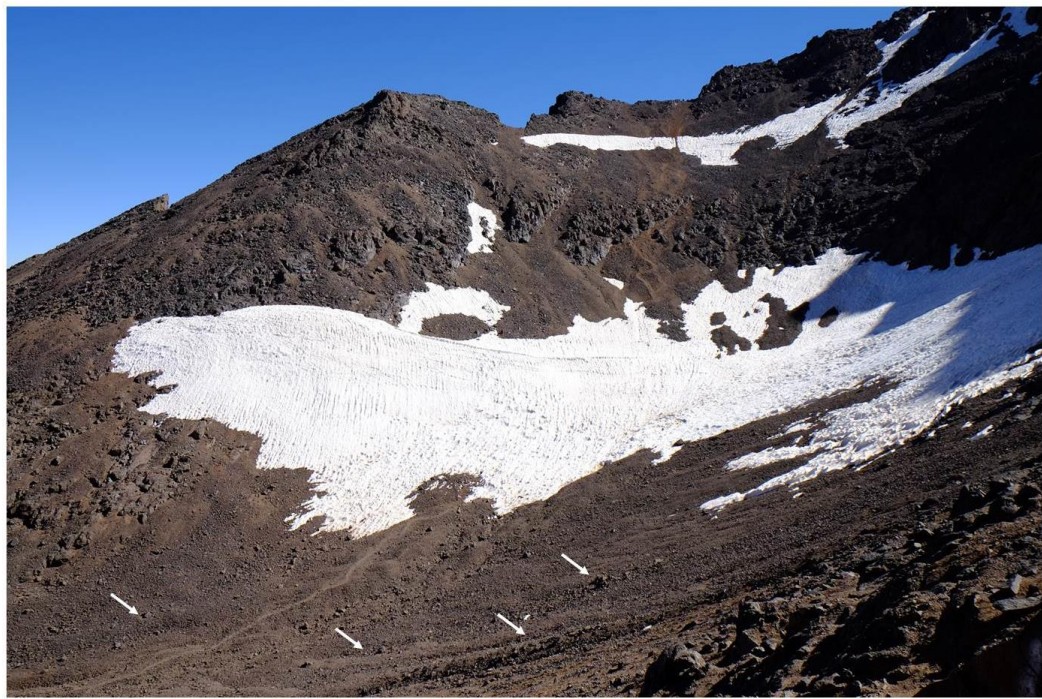

**Figure 12: The Irhzer Ikhibi south valley, where the datalogger Toubkal 3 was installed. The arrows indicate arcuate boulder ridges and furrows in the talus slope.**





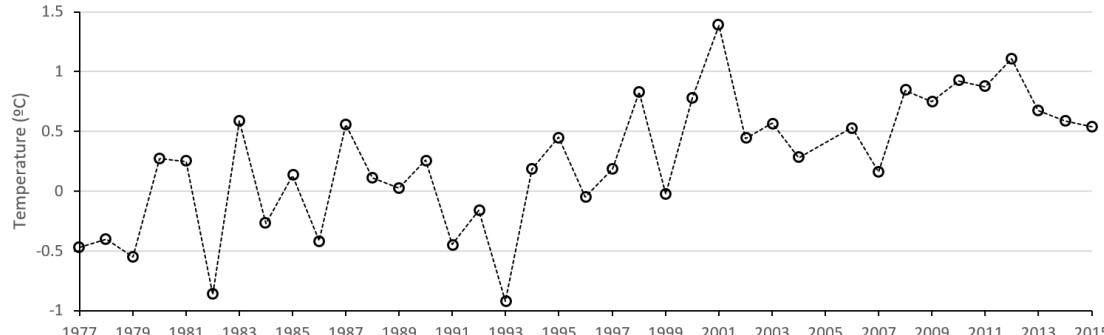

**Figure 13: Estimated mean annual air temperatures at the Djebel Toubkal (4,167 m asl) obtained from extrapolation of temperatures from Menara (Marrakesh) using a measured lapse rate of -0.59 ºC.100 m⁻¹.**