# Peer review of "New observations indicate the possible presence of permafrost in North Africa (Djebel Toubkal, High Atlas, Morocco)"

_The Cryosphere, 2016_

## Referee Comment (RC1) · Prof. Dr. E. Serrano (Referee) · 21 Dec 2016

The paper "Ground surface temperatures indicate the presence of permafrost in North Africa (Djebel Toubkal, High Atlas, Morocco)" is very interesting because it describes the possible presence of permafrost in the highest massif of northern Africa. The Atlas mountain permafrost has not been previously studied, so it is of interest to have a first approximation on permafrost in North Africa. This approach focuses on a small size area, although located in the highest massif of the Atlas, so it is significant to detect the possible presence of mountain permafrost. The paper is well structured and the figures clear and informative. The article presents a clear, simple and sufficiently contrasted methodology, which in my opinion is sufficiently effective to achieve the objectives, to

detect the possible presence of permafrost in the Toubkal massif. The comparison with the climatic conditions presents strong limitations, as the soil temperatures are compared with extrapolations. This is the weakest point, although with the above limitations, the work and the results are useful to compare the estimated environmental conditions with the soil records. The analysis of landforms is poor. A mapping of significant active periglacial landforms can support very good information, joint the snow permanence. Some active landforms are good indicators of permafrost or seasonal ice and have been used by numerous authors. In the text differences between lobate deposits and transverse ridges and furrows have been established. A greater accuracy on existing periglacial active landforms can allow the localization of frozen soils. Mapping and differentiation between landforms as gelifluction lobes, protalus lobes, frost mounds or rock glaciers, show places where are developed and where they are not developed, and they can permit extrapolate the frozen ground from the sites where data loggers have been located with the surroundings. The conclusions are in line with the data obtained and are relevant for the basic characterization of possible seasonal grounds or permafrost in the massif, offering the possibility of future mapping of permafrost in the massif. So, the paper is suitable for The Crysophere with very minor corrections.

From a formal perspective, several errata have been detected:

Pag. 2. line 31. Robinson and Williams, 1992, is not referred in the bibliography. Pag. 4, line 13. Cheggour, 2008, is not referenced in blibiography. Pag. 4, lines 26 and 30. The reference Chardon and Riser (1998) must be Chardon and Riser, 1981. Page 7, line 24. Figure 9, must be Figure 8. Page 10, line 7. Figure 12, must be Figure 11. Reference list: Oliva et al. 2016 is not cited in the text.

---

## Referee Comment (RC2) · B. Staub (Referee) · 7 Jan 2017

**GENERAL COMMENTS**

The research article "Ground surface temperatures indicate the presence of permafrost in North Africa (Djebel Toubkal, High Atlas, Morocco)" presents new observations and thoughts on the permafrost occurrence in the High Atlas. Due to the particularly poor data basis concerning permafrost and related phenomena in North Africa, any new measurements and findings on the potential permafrost distribution are of high importance, not only for the research community but also from an environmental and socio-economic perspective.

[Figure]

The gap in research and the knowledge from previous studies are well described in the introduction and the topic is nicely put into a larger context illustrating the characteristics of arid and semi-arid environments. Moreover, the environmental conditions and the geological setting of the studied region are compared with other high mountain areas. The overall objective, to contribute to the question of permafrost distribution in North Africa, is very ambitious. Probably too ambitious because recent studies and direct observations are missing and the acquired GST time series only cover one year of data for a small number of spots. Although the authors grade their work as an "exploratory step towards an in-depth assessment aiming at the characterization and modelling of permafrost in the High Atlas", the main critique on this article could be: Why didn't the authors wait two years longer to publish their work together with a sound data basis, consisting of at least three years of continuous GST data and e.g. some complementary geophysical investigations?

The general approach combining weather station data series, remote sensing data and a geomorphological interpretation of the landforms is certainly a good starting point to maximise the informative value of the GST data. However, although the authors clearly point out that the few observations must be interpreted very carefully, the measurement setup and some of the results seem somehow on the limit of being scientifically reasonable. Personally, I doubt that only four miniature loggers can provide a meaningful 'altitudinal gradient', and I would avoid making linear regressions (P7_L3 & Fig. 8) out of it nor extrapolate these results to a larger area. On the other hand, the publication can be justified because almost no permafrost observations are available for the High Atlas, and because this paper may motivate the mountain permafrost research community to put particular emphasis on that region. Moreover, the article is a nice complement to other mountain permafrost papers enriching the TC special issue "The evolution of permafrost in mountain regions". Therefore, I recommend this research article to be published with minor corrections. The following comments could help to improve the article, mainly concerning the interpretation of the GST data. Most important, I would like to encourage the authors to keep on measuring and observing the

permafrost in the High Atlas!

SPECIFIC COMMENTS

Title: The experience from the European Alps showed that one year of GST measurements does not provide reliable results on the ground thermal regime because of the high inter-annual variability of weather and snow conditions. In this regard, I suggest rethinking the title of the manuscript, e.g. towards a more neutral formulation "New observations indicate the occurrence of permafrost in the High Atlas mountain range (Djebel Toubkal, Morocco)".

Interpretation of GST data: The interpretation of the GST data as a 'BTS signal' is only valid if a thermally insulating snow cover is present for longer than just a few weeks. It seems like logger T3 fulfils this criterion around end of February 2016 (Fig. 10). At T1 and T2, the active layer is likely not in a thermal equilibrium with the permafrost base, these GST records characterise an integral of the recent atmospheric conditions with some modification by a temporarily snow cover. Depending on the terrain roughness and the snow density, about 50-100 cm of snow are required to effectively insulate the ground surface from air temperature variations (e.g. Keller and Gubler (1993); Zhang (2005); Staub and Delaloye (2016)). If there is less snow under winter conditions, the ground is likely colder at its surface than a few meters below. Although the potentially snow-covered period is shorter and snow heights are lower on average in the Toubkal massif than e.g. in the European Alps, the timing and duration of the snow cover probably play a key role for sporadic permafrost occurrence also in the High Atlas – as described by the authors. In comparison to the permafrost areas in the Alps, where the conditions during the winter season are often more important for inter-annual ground temperature variations than the summer warming (cf. PERMOS (2016)), snow disappearance is up to three months earlier in the Toubkal massif (despite of persisting snow patches), even at 3500 m asl. This means that the ground is usually snow free during the entire period of maximal insulation. The local effects of shading could be very important. Probably a GIS-analysis on topo-climatic parameters such as potential

incoming solar radiation, slope and curvature could help to characterise the acquired GST data and putting it into the spatial context. Moreover, it might be interesting to quantify ground thawing and freezing degree day sums for the summer and winter period.

Measurement setup: A future GST measurement setup around Djebel Toubkal could be installed similarly as described by Gubler et al. (2011) to provide some observational evidence on the GST variability considering different ground materials and topoclimatic situations at least for a few years. I am fully aware of the high financial and logistical effort for such permafrost observations in the remote High Atlas, but I think that such a data basis is required for any further steps towards permafrost mapping and modelling. At best, such spatially distributed GST measurements would be complemented by ERT surveys and geomorphological mapping. Building up a rock glacier and frozen debris lobe inventory

Weather and climate data: The authors characterise that particular year with GST observations in the climatological context by using meteorological data (Sect. 4.1, Figs. 3-5, 11 and 13). This is clearly a challenging task regarding the sparse data available, but the spatial transfer of air temperature data over ~3500 m elevation between Menara at Marrakesh to the Djebel Toubkal mountain is not satisfactory from a scientific point of view. Although the lapse rates provided and calculated seem plausible, these lapse rates likely vary over the season and the weather conditions can be very different in the mountains to what is measured in Marrakesh. The "significant correlation" of monthly air temperature values between Neltner and Sidi Chamarouch (P6_L14 and Fig. 3) is likely a result of the high seasonal temperature amplitudes. However, the comparison to other quantitative and qualitative data sources could be extended. For example, the snow climatology could be analysed over the entire period of available satellite imagery. Also satellite-derived land surface temperature data could enhance the comparison of the period 2015-16 in a larger temporal and spatial context – of course with limitations due to the lack of validation data and the difficulties in mountainous terrain. Maybe

even RCM reanalysis data could help to assess the regional climate history.

P7_L20: Clarify that you mean daily maxima in the sentence "A plateau in the maxima..."

P7_L23-24: I would not state the relationship between elevation and MAGST of these four locations as "statistically significant" and rather try to quantify the uncertainty of each data point. The uncertainty of MAGST is likely much higher than $\pm 0.4°C/100m$. Observations from the Swiss Alps show that elevation can be a poor proxy for MAGST, depending on the terrain and snow characteristics (boulder size, terrain roughness, solar irradiation, exposure to wind, and accumulation of snow by wind or avalanches) and regional weather patterns (e.g. Gubler et al (2011)).

Fig. 5: Clarify, that the dashed line is the extrapolation for the summit of Djebel Toubkal. If possible, add an uncertainty estimate (e.g. using a range of lapse rates).

Fig. 6: Add readable point labels and a legend for the colours.

Fig. 7: What are "daily hourly maxima"?

Fig. 8: See comment above. Maybe add an uncertainty estimate to each point?

Fig. 10: GST data series can be calibrated during the zero curtain period. It is visually not clear, if this calibration was done or if the dashed line is not really at 0°C at some of the time series.

REFERENCES

Gubler, S., Fiddes, J., Keller, M. and Gruber, S. (2011): Scale-dependent measurement and analysis of ground surface temperature variability in alpine terrain, The Cryosphere, 5(2), 431–443, doi:10.5194/tc-5-431-2011. Keller, F. and Gubler, H. (1993): Interaction between snow cover and high mountain permafrost, Murtèl-Corvatsch, Swiss Alps, in Proceedings of the 6th International Conference on Permafrost, Beijing, China, vol. 1, edited by J. Brown, H. M. French, N. A. Grave, C.

Guodong, L. King, E. A. Koster, and T. L. Pévé, pp. 332–337, South China University of Technology Press, Wushan Guangzhou China. PERMOS (2016): Permafrost in Switzerland 2010/2011 to 2013/2014. Noetzli, J. , Luethi, R., and Staub, B. (ed), Glaciological Report Permafrost No. 12–15 of the Cryospheric Commission of the Swiss Academy of Sciences, Fribourg, Switzerland. Staub, B. and Delaloye, R. (2016): Using Near-Surface Ground Temperature Data to Derive Snow Insulation and Melt Indices for Mountain Permafrost Applications, Permafrost and Periglacial Processes, doi:10.1002/ppp.1890. Zhang, T. (2005): Influence of the seasonal snow cover on the ground thermal regime: An overview, Reviews of Geophysics, 43(4), RG4002, doi:10.1029/2004RG000157.

---

## Author Comment (AC1) · 26 Jan 2017

Reply to Reviewer 1

Dear Prof Enrique Serrano,

Thank you very much for the review and interest in manuscript. Your comments indicate a small number of formal edits, which we will apply in the final form of the manuscript, as well as two points which are weakest in the manuscript :

Comment by E. Serrano : Âń The comparison with the climatic conditions presents strong limitations, as the soil temperatures are compared with extrapolations. This is the weakest point, although with the above limitations, the work and the results are

useful to compare the estimated environmental conditions with the soil records. Âż

Reply by the authors : We agree with your comments. However, given the scarce data available for the highest reaches of the High Atlas, we think that this was the best approach that could be done. In particular, our goal was to evaluate if the data from the year 2015-16 is representative of the climate of the area and how it fits with the interannual climate variability. The comparison with data from the lowlands in the north (Menara) shows statistically significant correlations and we use it to frame the study period in a longer period, but accounting only for the regional climatic scale. Surely, this approach has limitations, but such limitations are also clear for the reader, allowing for a straightforward evaluation of the quality and problems with our assumptions.

Comment by Prof. E. Serrano : Âń The analysis of landforms is poor. A mapping of significant active periglacial landforms can support very good information, joint the snow permanence. Some active landforms are good indicators of permafrost or seasonal ice and have been used by numerous authors. In the text differences between lobate deposits and transverse ridges and furrows have been established. A greater accuracy on existing periglacial active landforms can allow the localization of frozen soils. Âż

Reply by the authors : Your are correct. However, the focus of this manuscript has been on ground surface temperatures and we have decided only to briefly mention the geomorphological phenomena across the section which we have analysed. The periglacial landforms are very limited spatially in this area, relatively monotonous (dominated by scree slopes) and the added value of a small-scale topographical map would not change our conclusions. Such an approach would also require much more field work and a different scope. This study was essentially prospective and given the results we have obtained, a future larger project is envisaged for the area. This project only benefited from funding for travel expenses for a few days. This is also the reason for the incipient monitoring approach.

We hope you accept our replies and we will improve the manuscript by including the
errata indicated. We will follow your comments and clarify the above-mentioned issues in the text.

Thank you ver much and our best wishes,

Gonçalo Vieira

---

## Author Comment (AC2) · 26 Jan 2017

Comments posted by reviewer 2, Dr Benno Staub.

GENERAL COMMENTS

The research article "Ground surface temperatures indicate the presence of permafrost in North Africa (Djebel Toubkal, High Atlas, Morocco)" presents new observations and thoughts on the permafrost occurrence in the High Atlas. Due to the particularly poor data basis concerning permafrost and related phenomena in North Africa, any new measurements and findings on the potential permafrost distribution are of high importance, not only for the research community but also from an environmental and

socio-economic perspective.

The gap in research and the knowledge from previous studies are well described in the introduction and the topic is nicely put into a larger context illustrating the characteristics of arid and semi-arid environments. Moreover, the environmental conditions and the geological setting of the studied region are compared with other high mountain areas. The overall objective, to contribute to the question of permafrost distribution in North Africa, is very ambitious. Probably too ambitious because recent studies and direct observations are missing and the acquired GST time series only cover one year of data for a small number of spots. Although the authors grade their work as an "exploratory step towards an in-depth assessment aiming at the characterization and modelling of permafrost in the High Atlas", the main critique on this article could be: Why didn't the authors wait two years longer to publish their work together with a sound data basis, consisting of at least three years of continuous GST data and e.g. some complementary geophysical investigations?

The general approach combining weather station data series, remote sensing data and a geomorphological interpretation of the landforms is certainly a good starting point to maximise the informative value of the GST data. However, although the authors clearly point out that the few observations must be interpreted very carefully, the measurement setup and some of the results seem somehow on the limit of being scientifically reasonable. Personally, I doubt that only four miniature loggers can provide a meaningful 'altitudinal gradient', and I would avoid making linear regressions (P7_L3 & Fig. 8) out of it nor extrapolate these results to a larger area. On the other hand, the publication can be justified because almost no permafrost observations are available for the High Atlas, and because this paper may motivate the mountain permafrost research community to put particular emphasis on that region. Moreover, the article is a nice complement to other mountain permafrost papers enriching the TC special issue "The evolution of permafrost in mountain regions". Therefore, I recommend this research article to be published with minor corrections. The following comments could help to
improve the article, mainly concerning the interpretation of the GST data. Most important, I would like to encourage the authors to keep on measuring and observing the permafrost in the High Atlas!

SPECIFIC COMMENTS

Title: The experience from the European Alps showed that one year of GST measurements does not provide reliable results on the ground thermal regime because of the high inter-annual variability of weather and snow conditions. In this regard, I suggest rethinking the title of the manuscript, e.g. towards a more neutral formulation "New observations indicate the occurrence of permafrost in the High Atlas mountain range (Djebel Toubkal, Morocco)".

Interpretation of GST data: The interpretation of the GST data as a 'BTS signal' is only valid if a thermally insulating snow cover is present for longer than just a few weeks. It seems like logger T3 fulfils this criterion around end of February 2016 (Fig. 10). At T1 and T2, the active layer is likely not in a thermal equilibrium with the permafrost base, these GST records characterise an integral of the recent atmospheric conditions with some modification by a temporarily snow cover. Depending on the terrain roughness and the snow density, about 50-100 cm of snow are required to effectively insulate the ground surface from air temperature variations (e.g. Keller and Gubler (1993); Zhang (2005); Staub and Delaloye (2016)). If there is less snow under winter conditions, the ground is likely colder at its surface than a few meters below. Although the potentially snow-covered period is shorter and snow heights are lower on average in the Toubkal massif than e.g. in the European Alps, the timing and duration of the snow cover probably play a key role for sporadic permafrost occurrence also in the High Atlas – as described by the authors. In comparison to the permafrost areas in the Alps, where the conditions during the winter season are often more important for inter-annual ground temperature variations than the summer warming (cf. PERMOS (2016)), snow disappearance is up to three months earlier in the Toubkal massif (despite of persisting snow patches), even at 3500 m asl. This means that the ground is usually snow free

during the entire period of maximal insulation. The local effects of shading could be very important. Probably a GIS-analysis on topo-climatic parameters such as potential incoming solar radiation, slope and curvature could help to characterise the acquired GST data and putting it into the spatial context. Moreover, it might be interesting to quantify ground thawing and freezing degree day sums for the summer and winter period.

Measurement setup: A future GST measurement setup around Djebel Toubkal could be installed similarly as described by Gubler et al. (2011) to provide some obser-vational evidence on the GST variability considering different ground materials and topoclimatic situations at least for a few years. I am fully aware of the high financial and logistical effort for such permafrost observations in the remote High Atlas, but I think that such a data basis is required for any further steps towards permafrost map-ping and modelling. At best, such spatially distributed GST measurements would be complemented by ERT surveys and geomorphological mapping. Building up a rock glacier and frozen debris lobe inventory. Weather and climate data: The authors char-acterise that particular year with GST observations in the climatological context by using meteorological data (Sect. 4.1, Figs. 3-5, 11 and 13). This is clearly a challeng-ing task regarding the sparse data available, but the spatial transfer of air temperature data over âĹij3500 m elevation between Menara at Marrakesh to the Djebel Toubkal mountain is not satisfactory from a scientific point of view. Although the lapse rates provided and calculated seem plausible, these lapse rates likely vary over the season and the weather conditions can be very different in the mountains to what is measured in Marrakesh. The "significant correlation" of monthly air temperature values between Neltner and Sidi Chamarouch (P6_L14 and Fig. 3) is likely a result of the high seasonal temperature amplitudes. However, the comparison to other quantitative and qualitative data sources could be extended. For example, the snow climatology could be analysed over the entire period of available satellite imagery. Also satellite-derived land surface temperature data could enhance the comparison of the period 2015-16 in a larger tem-poral and spatial context – of course with limitations due to the lack of validation data

and the difficulties in mountainous terrain. Maybe c even RCM reanalysis data could help to assess the regional climate history.

P7_L20: Clarify that you mean daily maxima in the sentence "A plateau in the maxima. . .'"

P7_L23-24: I would not state the relationship between elevation and MAGST of these four locations as "statistically significant" and rather try to quantify the uncertainty of each data point. The uncertainty of MAGST is likely much higher than $\pm0.4$ÅŮęC/100m. Observations from the Swiss Alps show that elevation can be a poor proxy for MAGST, depending on the terrain and snow characteristics (boulder size, terrain roughness, solar irradiation, exposure to wind, and accumulation of snow by wind or avalanches) and regional weather patterns (e.g. Gubler et al (2011)).

Fig. 5: Clarify, that the dashed line is the extrapolation for the summit of Djebel Toubkal. If possible, add an uncertainty estimate (e.g. using a range of lapse rates).

Fig. 6: Add readable point labels and a legend for the colours.

Fig. 7: What are "daily hourly maxima"?

Fig. 8: See comment above. Maybe add an uncertainty estimate to each point?

Fig. 10: GST data series can be calibrated during the zero curtain period. It is visually not clear, if this calibration was done or if the dashed line is not really at 0ÅŮęC at some of the time series.

REFERENCES

Gubler, S., Fiddes, J., Keller, M. and Gruber, S. (2011): Scale-dependent measurement and analysis of ground surface temperature variability in alpine terrain, The Cryosphere, 5(2), 431–443, doi:10.5194/tc-5-431-2011.

Keller, F. and Gubler, H. (1993): Interaction between snow cover and high mountain permafrost, Murtèl Corvatsch, Swiss Alps, in Proceedings of the 6th International Conference on Permafrost, Beijing, China, vol. 1, edited by J. Brown, H. M. French, N. A. Grave, C. Guodong, L. King, E. A. Koster, and T. L. Pévé, pp. 332–337, South China University of Technology Press, Wushan Guangzhou China.

PERMOS (2016): Permafrost in Switzerland 2010/2011 to 2013/2014.

Noetzli, J. , Luethi, R., and Staub, B. (ed), Glaciological Report Permafrost No. 12–15 of the Cryospheric Commission of the Swiss Academy of Sciences, Fribourg, Switzerland.

Staub, B. and Delaloye, R. (2016): Using Near-Surface Ground Temperature Data to Derive Snow Insulation and Melt Indices for Mountain Permafrost Applications, Permafrost and Periglacial Processes, doi:10.1002/ppp.1890.

Zhang, T. (2005): Influence of the seasonal snow cover on the ground thermal regime: An overview, Reviews of Geophysics, 43(4), RG4002, doi:10.1029/2004RG000157.

\*\*\*\*\*\*

REPLY TO DR BENNO STAUB,

Dear Dr Benno Staub,

Thank you very much for your insightful and detailed review with very valuable and important comments. Below we have selected the various questions and remarks that you pose and we address them. We will submit the manuscript following your suggestions in the next few days.

Best wishes,

Gonçalo Vieira, Carla Mora and Ali Faleh

\*\*

Comment by Dr Benno Staub : Âń The overall objective, to contribute to the question of permafrost distribution in North Africa, is very ambitious.  Probably too ambitious

because recent studies and direct observations are missing and the acquired GST time series only cover one year of data for a small number of spots. Although the authors grade their work as an "exploratory step towards an in-depth assessment aiming at the characterization and modelling of permafrost in the High Atlas", the main critique on this article could be: Why didn't the authors wait two years longer to publish their work together with a sound data basis, consisting of at least three years of continuous GST data and e.g. some complementary geophysical investigations? Âń

Reply by the authors : we agree with this criticism, but this was the only solution we had available, especially following the possibility to promote bilateral cooperation Portugal-Morocco. The question of the possible presence of permafrost in the High Atlas has not really been addressed previously in the literature and only scattered observations are found, mainly mentioning periglacial landforms, as we explain in the literature review. This being so, attracting significant funding in a competitive call for a mid-term (3 year project) would be almost impossible, since it would only be sustained by the meagre literature and broad working hypotheses. The option was to do an exploratory approach and for that, the funding available were bilateral agreements that partially fund travelling expenses (a few days) and no equipment. The scarce funding, remoteness of the sites and lack of possibility to check on the instrumentation, limited the experimental design to the one we follow in the manuscript. If the hypothesis of permafrost presence was to be confirmed, sustained by peer-reviewed result publication, then a full project application could follow. This is the rationale on the base of our approach. Bilateral projects are 1+1 year (depending of results of year 1) and the funding cycle has driven the science produced. However, we clearly agree that a longer time series is needed and we expect to implement a much better network in the near-future. Comment by Dr Benno Staub : Âń ... However, although the authors clearly point out that the few observations must be interpreted very carefully, the measurement setup and some of the results seem somehow on the limit of being scientifically reasonable. Personally, I doubt that only four miniature loggers can provide a meaningful 'altitudinal gradient', and I would avoid making linear regressions (P7_L3 & Fig. 8) out of it nor extrapolate

these results to a larger area. On the other hand, the publication can be justified because almost no permafrost observations are available for the High Atlas, and because this paper may motivate the mountain permafrost research community to put particular emphasis on that region.

Reply by the authors : We understand the concerns of the reviewer, but we think that we didn't go beyond the scientific reasonability of our results. Across the manuscript we try to balance the fact that the area is almost unknown and not go too far in our conclusions. In what concerns to the comments relating to P7_L3 and Fig.8, the issues are different. In P7_L3 we present results on altitudinal lapse rates from monthly air temperatures. The values that we have obtained are close to the ones from other authors and can be useful also for comparison with other mountain ranges, as for example is presented in well-known synthesis such as Barry or Geiger on mountain and local climates. In Fig. 8, we agree that conditions with soil are very much dependent on micro and toposcale factors, but in our approach we have carefully selected the sites so that they could minimize such influences and maximize the influence of altitude on GST. The exception was T3, which despite the similar overall micro-scale conditions, was in a concave in a valley slope, where snow showed a prevailing influence. This fact is also responsible by the larger residuals. We will follow the suggestions indicated below in the specific comments in order to include uncertainty in the data analysis. We have now also found that we forgot to refer to Fig. 8 in that same paragraph.

SPECIFIC COMMENTS (BY DR BENNO STAUB)

Comment by Dr Benno Staub : Âń Title: The experience from the European Alps showed that one year of GST measurements does not provide reliable results on the ground thermal regime because of the high inter-annual variability of weather and snow conditions. In this regard, I suggest rethinking the title of the manuscript, e.g. towards a more neutral formulation "New observations indicate the occurrence of permafrost in the High Atlas mountain range (Djebel Toubkal, Morocco)". Âń

Reply by the authors : We agree and we will change them title as suggested.

Comment by Dr Benno Staub : Âń Interpretation of GST data: The interpretation of the GST data as a 'BTS signal' is only valid if a thermally insulating snow cover is present for longer than just a few weeks. It seems like logger T3 fulfils this criterion around end of February 2016 (Fig. 10). At T1 and T2, the active layer is likely not in a thermal equilibrium with the permafrost base, these GST records characterise an integral of the recent atmospheric conditions with some modification by a temporarily snow cover. Depending on the terrain roughness and the snow density, about 50-100 cm of snow are required to effectively insulate the ground surface from air temperature variations (e.g. Keller and Gubler (1993); Zhang (2005); Staub and Delaloye (2016)). If there is less snow under winter conditions, the ground is likely colder at its surface than a few meters below. Although the potentially snow-covered period is shorter and snow heights are lower on average in the Toubkal massif than e.g. in the European Alps, the timing and duration of the snow cover probably play a key role for sporadic permafrost occurrence also in the High Atlas – as described by the authors. In comparison to the permafrost areas in the Alps, where the conditions during the winter season are often more important for inter-annual ground temperature variations than the summer warming (cf. PERMOS (2016)), snow disappearance is up to three months earlier in the Toubkal massif (despite of persisting snow patches), even at 3500 m asl. This means that the ground is usually snow free during the entire period of maximal insulation. The local effects of shading could be very important. Probably a GIS-analysis on topo-climatic parameters such as potential incoming solar radiation, slope and curvature could help to characterise the acquired GST data and putting it into the spatial context. Moreover, it might be interesting to quantify ground thawing and freezing degree day sums for the summer and winter period. Âż

Reply by the authors : This is an issue which was well thought during the preparation of the manuscript and we have decided to leave it as we present it. We have weighted well the terminology in order to be objective and stick to the data and to minimize
interpretations. Data shows that T3 is the only site where the Âń BTS Âż assumptions are valid and is the only site where data supports the occurrence of permafrost. For the highest sites, there is no indication that permafrost is present and probably it is not. The shadow effect and also the snow cover and its timing should be the key factors conditioning permafrost distribution in the High Atlas. We thought about using a GIS-based radiation modelling approach, but the points are few and not variable enough (e.g. T1 and T2 are in ridge position with comparable potential radiation, while T3 will have less radiation, but there is no real reason to sustain such a quantitative approach based on potential radiation, with this small sampling sites). The freezing and thawing degree days have been calculated for all sites and we have discussed within the team and with other specialists on their inclusion in the manuscript. However, this indexes were derived essentially for the Polar latitudes and we have decided to stick with the observed data. If needed, we can easily accomodate them in the manuscript, but we would prefer to use this, together with the empirico-statistical modelling approach in a forthcoming study with a much larger number of miniloggers.

Comment by Dr Benno Staub : Âń Measurement setup: A future GST measurement setup around Djebel Toubkal could be installed similarly as described by Gubler et al. (2011) to provide some observational evidence on the GST variability considering different ground materials and topoclimatic situations at least for a few years. I am fully aware of the high financial and logistical effort for such permafrost observations in the remote High Atlas, but I think that such a data basis is required for any further steps towards permafrost mapping and modelling. At best, such spatially distributed GST measurements would be complemented by ERT surveys and geomorphological mapping. Building up a rock glacier and frozen debris lobe inventory. Âż

Reply by the authors : You are right. The setup by Gubler et al (2011) is well-known by our team and we don't really know how we have missed it in the state of the art. It will surely ne one of the experimental setups that we will follow, together with a better altitudinal and aspect design. We will make sure we will accomodate the reference to

Gubler et al. (2011) in the manuscript.

Comment by Dr Benno Staub : Âń Weather and climate data: The authors characterise that particular year with GST observations in the climatological context by using meteorological data (Sect. 4.1, Figs. 3-5, 11 and 13). This is clearly a challenging task regarding the sparse data available, but the spatial transfer of air temperature data over âĹij3500 m elevation between Menara at Marrakesh to the Djebel Toubkal mountain is not satisfactory from a scientific point of view. Although the lapse rates provided and calculated seem plausible, these lapse rates likely vary over the season and the weather conditions can be very different in the mountains to what is measured in Marrakesh. The "significant correlation" of monthly air temperature values between Neltner and Sidi Chamarouch (P6_L14 and Fig. 3) is likely a result of the high seasonal temperature amplitudes. However, the comparison to other quantitative and qualitative data sources could be extended. For example, the snow climatology could be analysed over the entire period of available satellite imagery. Also satellite-derived land surface temperature data could enhance the comparison of the period 2015-16 in a larger temporal and spatial context – of course with limitations due to the lack of validation data and the difficulties in mountainous terrain. Maybe c even RCM reanalysis data could help to assess the regional climate history.

Reply by the authors : we agree with your comments. We will approach the regional climate variability using grided reanalysis data and evaluate how it improves the current approach. Snow climatology from remote sensing data would be the scope of a whole new approach, especially due to winter cloudiness and we will not do it here. The same applies to Land surface temperatures which are very much dependent on cloudiness and time of the day and there is not too much validation data. We will improve this part and change it in the next few days.

P7_L20: Clarify that you mean daily maxima in the sentence "A plateau in the maxima. . ."

Reply: Right. We will change it to ¿A plateau in the daily maxima... ¿

P7_L23-24: I would not state the relationship between elevation and MAGST of these four locations as "statistically significant" and rather try to quantify the uncertainty of each data point. The uncertainty of MAGST is likely much higher than ±0.4ŮȩC/100m. Observations from the Swiss Alps show that elevation can be a poor proxy for MAGST, depending on the terrain and snow characteristics (boulder size, terrain roughness, solar irradiation, exposure to wind, and accumulation of snow by wind or avalanches) and regional weather patterns (e.g. Gubler et al (2011)).

Reply : we will rephrase this sentence and will review this approach. The sites were selected in order to have a larger effect of altitude than of other factors, but including uncertainty in the analysis is an important point.

Fig. 5: Clarify, that the dashed line is the extrapolation for the summit of Djebel Toubkal. If possible, add an uncertainty estimate (e.g. using a range of lapse rates).

Reply : OK, we will do that. Somehow it is missing in the legend.

Fig. 6: Add readable point labels and a legend for the colours.

Reply : OK, we will do as suggested.

Fig. 7: What are "daily hourly maxima"?

Reply : the caption will be corrected : Âń Extremes are absolute monthly maximum and minimum temperatures.

Fig. 8: See comment above. Maybe add an uncertainty estimate to each point?

Reply : Yes, we will do as suggested.

Fig. 10: GST data series can be calibrated during the zero curtain period. It is visually not clear, if this calibration was done or if the dashed line is not really at 0ŮȩC at some of the time series.

[Figure]

Reply : The calibration was not done, but in fact when preparing the figure, somehow the dashed line moved away from 0°C in some of the figures. We will correct this.

---

## Author Response (AR1)

The paper "Ground surface temperatures indicate the presence of permafrost in North Africa (Djebel Toubkal, High Atlas, Morocco)" is very interesting because it describes the possible presence of permafrost in the highest massif of northern Africa. The Atlas mountain permafrost has not been previously studied, so it is of interest to have a first approximation on permafrost in North Africa. This approach focuses on a small size area, although located in the highest massif of the Atlas, so it is significant to detect the possible presence of mountain permafrost. The paper is well structured and the figures clear and informative. The article presents a clear, simple and sufficiently contrasted methodology, which in my opinion is sufficiently effective to achieve the objectives, to detect the possible presence of permafrost in the Toubkal massif. The comparison with the climatic conditions presents strong limitations, as the soil temperatures are compared with extrapolations. This is the weakest point, although with the above limitations, the work and the results are useful to compare the estimated environmental conditions with the soil records. The analysis of landforms is poor. A mapping of significant active periglacial landforms can support very good information, joint the snow permanence. Some active landforms are good indicators of permafrost or seasonal ice and have been used by numerous authors. In the text differences between lobate deposits and transverse ridges and furrows have been established. A greater accuracy on existing periglacial active landforms can allow the localization of frozen soils. Mapping and differentiation between landforms as gelifluction lobes, protalus lobes, frost mounds or rock glaciers, show places where are developed and where they are not developed, and they can permit extrapolate the frozen ground from the sites where data loggers have been located with the surroundings. The conclusions are in line with the data obtained and are relevant for the basic characterization of possible seasonal grounds or permafrost in the massif, offering the possibility of future mapping of permafrost in the massif. So, the paper is suitable for The Crysophere with very minor corrections.

From a formal perspective, several errata have been detected:

Pag. 2. line 31. Robinson and Williams, 1992, is not referred in the bibliography.

Pag. 4, line 13. Cheggour, 2008, is not referenced in blibiography.

Pag. 4, lines 26 and 30. The reference Chardon and Riser (1998) must be Chardon and Riser, 1981.

Page 7, line 24. Figure 9, must be Figure 8.

Page 10, line 7. Figure 12, must be Figure 11.

Reference list: Oliva et al. 2016 is not cited in the text.

**REPLY BY AUTHORS SUBMITTED IN 6 MARCH 2017**

Dear Prof Enrique Serrano,

Thank you very much for the review and interest in manuscript. Your comments indicate a small number of formal edits, which we have now applied in the manuscript.

You also indicate two points which are weakest in the manuscript :

Comment by E. Serrano : « The comparison with the climatic conditions presents strong limitations, as the soil temperatures are compared with extrapolations. This is the weakest point, although with the above limitations, the work and the results are useful to compare the estimated environmental conditions with the soil records. »

> Reply by the authors : We agree with your comments. However, given the scarce data available for the highest reaches of the High Atlas, we think that this was the best approach that could be done. In particular, our goal was to evaluate if the data from the year 2015-16 is representative of the climate of the area and how it fits with the interannual climate variability. The comparison with data from the lowlands in the north (Menara) is used it to frame the study period in a longer period, but accounting only for the regional climatic scale. Surely, this approach has limitations, but such limitations are also clear for the reader, allowing for a straightforward evaluation of the quality and problems with our assumptions. We have also done several changes in the manuscript in order to limit the extrapolations to the strictly necessary.

Comment by Prof. E. Serrano : « The analysis of landforms is poor. A mapping of significant active periglacial landforms can support very good information, joint the snow permanence. Some active landforms are good indicators of permafrost or seasonal ice and have been used by numerous authors. In the text differences between lobate deposits and transverse ridges and furrows have been established. A greater accuracy on existing periglacial active landforms can allow the localization of frozen soils. »

> Reply by the authors : Your are correct. However, the focus of this manuscript has been solely on ground surface temperatures and we have decided only to briefly mention the geomorphological phenomena across the area which we have analysed. The periglacial landforms are very limited spatially in this area and the added value based on a small-scale topographical map would be small. Such an approach would also require much more field work and a different scope. This study was essentially prospective and given the results we have obtained, a future larger project is envisaged for the area. The current project only benefited from funding for travel expenses for a few days. This is also the reason for the incipient monitoring approach.

From a formal perspective, several errata have been detected:

Pag. 2. line 31. Robinson and Williams, 1992, is not referred in the bibliography.

> R : Added as indicated.

Pag. 4, line 13. Cheggour, 2008, is not referenced in blibiography.

> R : Added as indicated.

Pag. 4, lines 26 and 30. The reference Chardon and Riser (1998) must be Chardon and Riser, 1981.

R : Corrected as indicated

Page 7, line 24. Figure 9, must be Figure 8.

R : Corrected as indicated.

Page 10, line 7. Figure 12, must be Figure 11.

R : The figure number was OK.

Reference list: Oliva et al. 2016 is not cited in the text.

R : Now added in p.2, line 26.

We hope you accept our replies and that you find the mansucript acceptable in the new version.

Thank you very much and our best wishes,

Gonçalo Vieira, Carla Mora and Ali Faleh

**COMMENTS POSTED BY REVIEWER 2, DR BENNO STAUB.**

GENERAL COMMENTS

The research article "Ground surface temperatures indicate the presence of permafrost in North Africa (Djebel Toubkal, High Atlas, Morocco)" presents new observations and thoughts on the permafrost occurrence in the High Atlas. Due to the particularly poor data basis concerning permafrost and related phenomena in North Africa, any new measurements and findings on the potential permafrost distribution are of high importance, not only for the research community but also from an environmental and socio-economic perspective.

The gap in research and the knowledge from previous studies are well described in the introduction and the topic is nicely put into a larger context illustrating the characteristics of arid and semi-arid environments. Moreover, the environmental conditions and the geological setting of the studied region are compared with other high mountain areas. The overall objective, to contribute to the question of permafrost distribution in North Africa, is very ambitious. Probably too ambitious because recent studies and direct observations are missing and the acquired GST time series only cover one year of data for a small number of spots. Although the authors grade their work as an "exploratory step towards an in-depth assessment aiming at the characterization and modelling of permafrost in the High Atlas", the main critique on this article could be: Why didn't the authors wait two years longer to publish their work together with a sound data basis, consisting of at least three years of continuous GST data and e.g. some complementary geophysical investigations?

The general approach combining weather station data series, remote sensing data and a geomorphological interpretation of the landforms is certainly a good starting point to maximise the informative value of the GST data. However, although the authors clearly point out that the few observations must be interpreted very carefully, the measurement setup and some of the results seem somehow on the limit of being scientifically reasonable. Personally, I doubt that only four miniature loggers can provide a meaningful 'altitudinal gradient', and I would avoid making linear regressions (P7_L3 & Fig. 8) out of it nor extrapolate these results to a larger area. On the other hand, the publication can be justified because almost no permafrost observations are available for the High Atlas, and because this paper may motivate the mountain permafrost research community to put particular emphasis on that region. Moreover, the article is a nice complement to other mountain permafrost papers enriching the TC special issue "The evolution of permafrost in mountain regions". Therefore, I recommend this research article to be published with minor corrections. The following comments could help to improve the article, mainly concerning the interpretation of the GST data. Most important, I would like to encourage the authors to keep on measuring and observing the permafrost in the High Atlas!

SPECIFIC COMMENTS

Title: The experience from the European Alps showed that one year of GST measurements does not provide reliable results on the ground thermal regime because of the high inter-annual variability of weather and snow conditions. In this regard, I suggest rethinking the title of the manuscript, e.g. towards a more neutral formulation "New observations indicate the occurrence of permafrost in the High Atlas mountain range (Djebel Toubkal, Morocco)".

Interpretation of GST data: The interpretation of the GST data as a 'BTS signal' is only valid if a thermally insulating snow cover is present for longer than just a few weeks. It seems like logger T3 fulfils this criterion around end of February 2016 (Fig. 10). At T1 and T2, the active layer is likely not in a thermal equilibrium with the permafrost base, these GST records characterise an integral of the recent atmospheric conditions with some modification by a temporarily snow cover. Depending on the terrain roughness and the snow density, about 50-100 cm of snow are required to effectively insulate the ground surface from air temperature variations (e.g. Keller and Gubler (1993); Zhang (2005); Staub and Delaloye (2016)). If there is less snow under winter conditions, the ground is likely colder at its surface than a few meters below. Although the potentially snow-covered period is shorter and snow heights are lower on average in the Toubkal massif than e.g. in the European Alps, the timing and duration of the snow cover probably play a key role for sporadic permafrost occurrence also in the High Atlas – as described by the authors. In comparison to the permafrost areas in the Alps, where the conditions during the winter season are often more important for inter-annual ground temperature variations than the summer warming (cf. PERMOS (2016)), snow disappearance is up to three months earlier in the Toubkal massif (despite of persisting snow patches), even at 3500 m asl. This means that the ground is usually snow free during the entire period of maximal insulation. The local effects of shading could be very important. Probably a GIS-analysis on topo-climatic parameters such as potential incoming solar radiation, slope and curvature could help to characterise the acquired GST data and putting it into the spatial context. Moreover, it might be interesting to quantify ground thawing and freezing degree day sums for the summer and winter period.

Measurement setup: A future GST measurement setup around Djebel Toubkal could be installed similarly as described by Gubler et al. (2011) to provide some observational evidence on the GST variability considering different ground materials and topoclimatic situations at least for a few years. I am fully aware of the high financial and logistical effort for such permafrost observations in the remote High Atlas, but I think that such a data basis is required for any further steps towards permafrost mapping and modelling. At best, such spatially distributed GST measurements would be complemented by ERT surveys and geomorphological mapping. Building up a rock glacier and frozen debris lobe inventory.

Weather and climate data: The authors characterise that particular year with GST observations in the climatological context by using meteorological data (Sect. 4.1, Figs. 3-5, 11 and 13). This is clearly a challenging task regarding the sparse data available, but the spatial transfer of air temperature data over ~3500 m elevation between Menara at Marrakesh to the Djebel Toubkal mountain is not satisfactory from a scientific point of view. Although the lapse rates provided and calculated seem plausible, these lapse rates likely vary over the season and the weather conditions can be very different in the mountains to what is measured in Marrakesh. The "significant correlation" of monthly air temperature values between Neltner and Sidi Chamarouch (P6_L14 and Fig. 3) is likely a result of the high seasonal temperature amplitudes. However, the comparison to other quantitative and qualitative data sources could be extended. For example, the snow climatology could be analysed over the entire period of available satellite imagery. Also satellite-derived land surface temperature data could enhance the comparison of the period 2015-16 in a larger temporal and spatial context – of course with limitations due to the lack of validation data and the difficulties in mountainous terrain. Maybe c even RCM reanalysis data could help to assess the regional climate history.

P7_L20: Clarify that you mean daily maxima in the sentence "A plateau in the maxima. . ."

P7_L23-24: I would not state the relationship between elevation and MAGST of these four locations as "statistically significant" and rather try to quantify the uncertainty of each data point. The uncertainty of MAGST is likely much higher than ±0.4◦C/100m. Observations from the Swiss Alps show that elevation can be a poor proxy for MAGST, depending on the terrain and snow characteristics (boulder size, terrain roughness, solar irradiation, exposure to wind, and accumulation of snow by wind or avalanches) and regional weather patterns (e.g. Gubler et al (2011)).

Fig. 5: Clarify, that the dashed line is the extrapolation for the summit of Djebel Toubkal. If possible, add an uncertainty estimate (e.g. using a range of lapse rates).

Fig. 6: Add readable point labels and a legend for the colours.

Fig. 7: What are "daily hourly maxima"?

Fig. 8: See comment above. Maybe add an uncertainty estimate to each point?

Fig. 10: GST data series can be calibrated during the zero curtain period. It is visually not clear, if this calibration was done or if the dashed line is not really at 0◦C at some of the time series.

Dear Dr Benno Staub,

Thank you very much for your insightful and detailed review with very valuable and important comments. Below we have selected the various questions and remarks that you pose and we address them. The revised manuscript is also now submitted to the journal.

Best wishes,

Gonçalo Vieira, Carla Mora and Ali Faleh

Comment by Dr Benno Staub : « The overall objective, to contribute to the question of permafrost distribution in North Africa, is very ambitious. Probably too ambitious because recent studies and direct observations are missing and the acquired GST time series only cover one year of data for a small number of spots. Although the authors grade their work as an "exploratory step towards an in-depth assessment aiming at the characterization and modelling of permafrost in the High Atlas", the main critique on this article could be: Why didn't the authors wait two years longer to publish their work together with a sound data basis, consisting of at least three years of continuous GST data and e.g. some complementary geophysical investigations? «

> Reply by authors : we agree with this criticism, but this was the only solution we had available, especially following the possibility to promote bilateral cooperation Portugal-Morocco. The question of the possible presence of permafrost in the High Atlas has not really been addressed previously in the literature and only scattered observations are found, mainly mentioning periglacial landforms, as we explain in the literature review. This being so, attracting significant funding in a competitive call for a mid-term (3 year project) would be almost impossible, since it would only be sustained by the meagre literature and broad working hypotheses. The option was to do an exploratory approach and for that, the funding available were bilateral agreements that partially fund travelling expenses (a few days) and no equipment. The scarce funding, remoteness of the sites and lack of possibility to check on the instrumentation, limited the experimental design to the one we follow in the manuscript. If the hypothesis of permafrost presence was to be confirmed, sustained by peer-reviewed result publication, then a full project application could follow. This is the rationale on the base of our approach. Bilateral projects are 1+1 year (depending of results of year 1) and the funding cycle has driven the science produced. However, we clearly agree that a longer time series is needed and we expect to implement a much better network in the near-future.

Comment by Dr Benno Staub : « ... However, although the authors clearly point out that the few observations must be interpreted very carefully, the measurement setup and some of the results seem somehow on the limit of being scientifically reasonable. Personally, I doubt that only four miniature loggers can provide a meaningful 'altitudinal gradient', and I would avoid making linear regressions (P7_L3 & Fig. 8) out of it nor extrapolate these results to a larger area. On the other hand, the publication can be justified because almost no permafrost

observations are available for the High Atlas, and because this paper may motivate the mountain permafrost research community to put particular emphasis on that region.

> Reply by the authors : We understand the concerns of the reviewer, but we think that we didn't go beyond the scientific reasonability of our results. Across the manuscript we try to balance the fact that the area is almost unknown and not go too far in our conclusions. In what concerns to the comments relating to P7_L3 and Fig.8, the issues are different. In P7_L3 we present results on altitudinal lapse rates from monthly air temperatures. The values that we have obtained are close to the ones from other authors and can be useful also for comparison with other mountain ranges, as for example is presented in well-known synthesis such as Barry or Geiger on mountain and local climates. In Fig. 8, we agree that conditions with soil are very much dependent on micro and toposcale factors, but in our approach we have carefully selected the sites so that they could minimize such influences and maximize the influence of altitude on GST. The exception was T3, which despite the similar overall micro-scale conditions, was in a concave in a valley slope, where snow showed a prevailing influence. This fact is also responsible by the larger residuals. We have followed the suggestions and after thinking it thoroughly, we have decided to fully remove the correlation and best-fit and not even to include uncertainty measures. We simplified the analysis and kept to a description of the results.

SPECIFIC COMMENTS (BY DR BENNO STAUB)

Comment by Dr Benno Staub : « Title: The experience from the European Alps showed that one year of GST measurements does not provide reliable results on the ground thermal regime because of the high inter-annual variability of weather and snow conditions. In this regard, I suggest rethinking the title of the manuscript, e.g. towards a more neutral formulation "New observations indicate the occurrence of permafrost in the High Atlas mountain range (Djebel Toubkal, Morocco)". «

> Reply by the authors : We agree and we have changed the title as suggested.

Comment by Dr Benno Staub : « Interpretation of GST data: The interpretation of the GST data as a 'BTS signal' is only valid if a thermally insulating snow cover is present for longer than just a few weeks. It seems like logger T3 fulfils this criterion around end of February 2016 (Fig. 10). At T1 and T2, the active layer is likely not in a thermal equilibrium with the permafrost base, these GST records characterise an integral of the recent atmospheric conditions with some modification by a temporarily snow cover. Depending on the terrain roughness and the snow density, about 50-100 cm of snow are required to effectively insulate the ground surface from air temperature variations (e.g. Keller and Gubler (1993); Zhang (2005); Staub and Delaloye (2016)). If there is less snow under winter conditions, the ground is likely colder at its surface than a few meters below. Although the potentially snow-covered period is shorter and snow heights are lower on average in the Toubkal massif than e.g. in the European Alps, the timing and duration of the snow cover probably play a key role for sporadic permafrost occurrence also in the High Atlas – as described by the authors. In comparison to the permafrost areas in the Alps, where the conditions during the winter season are often more important for inter-annual ground temperature variations than the summer warming (cf. PERMOS (2016)), snow disappearance is up to three months earlier in the Toubkal massif (despite of persisting snow patches), even at 3500 m asl. This means that the ground is usually snow free during the entire

period of maximal insulation. The local effects of shading could be very important. Probably a GIS-analysis on topo-climatic parameters such as potential incoming solar radiation, slope and curvature could help to characterise the acquired GST data and putting it into the spatial context. Moreover, it might be interesting to quantify ground thawing and freezing degree day sums for the summer and winter period. »

> Reply by the authors : This is an issue which was well thought during the preparation of the manuscript and we have decided to leave it as we present it. We have weighted well the terminology in order to be objective and stick to the data and to minimize interpretations. Data shows that T3 is the only site where the « BTS » assumptions are valid and is the only site where data supports the occurrence of permafrost. For the highest sites, there is no indication that permafrost is present and probably it is not. The shadow effect and also the snow cover and its timing should be the key factors conditioning permafrost distribution in the High Atlas. We thought about using a GIS-based radiation modelling approach, but the points are few and not variable enough (e.g. T1 and T2 are in ridge position with comparable potential radiation, while T3 will have less radiation, but there is no real reason to sustain such a quantitative approach based on potential radiation, with this small sampling sites). The freezing and thawing degree days have been calculated for all sites and we have discussed within the team and with other specialists on their inclusion in the manuscript. However, this indexes were derived essentially for the Polar latitudes and we have decided to stick with the observed data. If needed, we can easily accomodate them in the manuscript, but we would prefer to use this, together with the empirico-statistical modelling approach in a forthcoming study with a much larger number of miniloggers.

Comment by Dr Benno Staub : « Measurement setup: A future GST measurement setup around Djebel Toubkal could be installed similarly as described by Gubler et al. (2011) to provide some observational evidence on the GST variability considering different ground materials and topoclimatic situations at least for a few years. I am fully aware of the high financial and logistical effort for such permafrost observations in the remote High Atlas, but I think that such a data basis is required for any further steps towards permafrost mapping and modelling. At best, such spatially distributed GST measurements would be complemented by ERT surveys and geomorphological mapping. Building up a rock glacier and frozen debris lobe inventory. »

> Reply by the authors : You are right. The setup by Gubler et al (2011) is well-known by our team and we don't really know how we have missed it in the state of the art. It will surely be one of the experimental setups that we will follow, together with a better altitudinal and aspect design. We have included a reference to Gubler et al. (2011) and wrote a sentence re-emphasising on the care needed for data interpretation on p. 5, lines 20-22.

Comment by Dr Benno Staub : « Weather and climate data: The authors characterise that particular year with GST observations in the climatological context by using meteorological data (Sect. 4.1, Figs. 3-5, 11 and 13). This is clearly a challenging task regarding the sparse data available, but the spatial transfer of air temperature data over ~3500 m elevation between Menara at Marrakesh to the Djebel Toubkal mountain is not satisfactory from a scientific point of view. Although the lapse rates provided and calculated seem plausible, these lapse rates likely vary over the season and the weather conditions can be very different in the mountains to what is measured in Marrakesh. The "significant correlation" of monthly air temperature

values between Neltner and Sidi Chamarouch (P6_L14 and Fig. 3) is likely a result of the high seasonal temperature amplitudes. However, the comparison to other quantitative and qualitative data sources could be extended. For example, the snow climatology could be analysed over the entire period of available satellite imagery. Also satellite-derived land surface temperature data could enhance the comparison of the period 2015-16 in a larger temporal and spatial context – of course with limitations due to the lack of validation data and the difficulties in mountainous terrain. Maybe c even RCM reanalysis data could help to assess the regional climate history.

> Reply by the authors : we agree with your comments related to the use of correlations with Menara and we removed that part from the manuscript. Figure 3 was deleted. We rewrote the methodology explaining that Menara is the only climate station with a long data series in the region allowing to assess on the climatic representativity of the study period. This means that we assume that Menara reflects the overall regional climate characteristics of warm vs cold months, and dry vs wet months. We dismissed using RCM reanalysis data since the grid size would impose a number of constraints, the approach would bring a new focus to the manuscript and the improvement would not be significant, especially since we are targeting at only a small number of GST loggers. We have preferred to concentrate on the data we have collected rather than on extrapolations and thus we have simplified the text by improving the description of the air temperature data measured at our sites (p.6, l29 to p.7, l4). Using snow climatology from remote sensing data would also be a whole new approach, especially due to winter cloudiness and we decided not to apply it here. The same applies to Land surface temperatures, which are very much dependent on cloudiness and time of the day and there is not too much validation data.

P7_L20: Clarify that you mean daily maxima in the sentence "A plateau in the maxima. . ."

> Reply: Right. We will change it to » plateau in the curve for the daily maxima … »

P7_L23-24: I would not state the relationship between elevation and MAGST of these four locations as "statistically significant" and rather try to quantify the uncertainty of each data point. The uncertainty of MAGST is likely much higher than ±0.4∘C/100m. Observations from the Swiss Alps show that elevation can be a poor proxy for MAGST, depending on the terrain and snow characteristics (boulder size, terrain roughness, solar irradiation, exposure to wind, and accumulation of snow by wind or avalanches) and regional weather patterns (e.g. Gubler et al (2011)).

> Reply : we have fully rewritten the sentence and simplified it in order to avoid misinterpretation relating to the best-fit line (p.7, l23-24). The best-fit was also removed from the figure and we stick to presenting the data.

Fig. 5 [NOW FIGURE 4]: Clarify, that the dashed line is the extrapolation for the summit of Djebel Toubkal. If possible, add an uncertainty estimate (e.g. using a range of lapse rates).

> Reply : We have decided to remove the extrapolation to the summit of the Toubkal.

Fig. 6 [NOW FIGURE 5]: Add readable point labels and a legend for the colours.

> Reply : OK, done as suggested.

Fig. 7 [NOW FIGURE 6]: What are "daily hourly maxima"?

Reply : the caption was corrected : « Extremes are absolute monthly maximum and minimum temperatures.

Fig. 8 [NOW FIGURE 7]: See comment above. Maybe add an uncertainty estimate to each point?

Reply : We have finally decided to remove the best-fit and therefore not to include the uncertainty.

Fig. 10 [NOW FIGURE 9]: GST data series can be calibrated during the zero curtain period. It is visually not clear, if this calibration was done or if the dashed line is not really at 0◦C at some of the time series.

Reply : The position of the dashed line is now correct. Somehow it had shifted during the editing process in the previous version.

We have added the following references :

[revised manuscript text omitted]
)" to The Cryosphere special issue on The evolution of permafrost in mountain regions. We have followed most of the reviewer's suggestions and made several changes to the text, improved the figures and removed figure 3. We have specially focused on limiting the component associated with climate extrapolations based on Menara data, but we kept those which were important for better framing the GST data. For exampleWe would like to point out that in the previous answer to Dr Benno Staub, we indicated that we would approach the retrieval of a regional climate time-series by using Reanalysis, but finally, we have decided that such an approach would not bring added value to the use of the data from Menara (Marrakesh) and we did not implement it.

The document with replies to the referees include first the full comments by the referee and then our detailed replies.

We think that the manuscript is now more solid and expect that it can be accepted for publication in The Cryosphere.

Thank you very much.

 Best wishes,

Gonçalo Vieira, Carla Mora and Ali Faleh

---

## Editor Decision (ED1)

**Comments to the Authors** on 'New observations indicate the presence of permafrost in North Africa (Djebel Toubkal, High Atlas, Morocco)' by G. Vieira, C. Mora and A. Faleh.

Dear Authors,

The paper has been improved but I feel it requires some major revisions for the following reasons. The two reviewers proposed a series of useful revisions, many of which were not carried out – for example with the rather unconvincing argument that 'a larger project is envisaged in future'. As the data basis presented is extremely meagre and was measured over a very short time period, as was underlined by Reviewer 2, I would have expected more of the Reviewers' suggestions to have been carried out to substantiate the existing observations and data – and to underline that future research would be of relevance in this area of North Africa.

Prof. E. Serrano (Reviewer 1) commented that the analysis of the landforms was poor - and although you describe some of them, it is impossible for the reader to locate these, as there is no overview of them. A simple geomorphological map, an aerial photograph or even an excerpt from Google Earth with labelled landforms would be useful. This is particularly important in this setting, because the distribution of possible permafrost does not appear to be linked to an elevational gradient – but rather to morphological terrain characteristics. The process of ice genesis at the base of steep slopes due to mass wasting (avalanches, rockfall) needs to be discussed further and will be mentioned again below.

Dr. B. Staub (Reviewer 2) suggested carrying out a GIS analysis to investigate potential solar radiation, slope angles and terrain curvature. This would allow interesting comparisons with GIS analyses carried out in other mountain regions, e.g. (Kenner and Magnusson, 2017) and would help to determine whether the presence of permafrost is possible or not. From a purely morphological point of view, a simple GIS analysis would allow to determine locations where snow avalanches and rock fall can be deposited – thus pointing towards potential locations of buried ice at the foot of slopes (where air temperature regimes, solar radiation and GST have less influence on the distribution of permafrost than the presence of massive ice buried under a layer of rock debris).

Reviewer 2 also suggested including freezing and thawing degree-days – this would be a further interesting way of comparing the GST regime at the 4 sites and of making first steps towards the development of High Atlas-specific GST indices – if they have any relation to permafrost distribution at all here. You state that you have these values and I suggest you include them. The extremely high summer GSTs at site T3 and the low ones in winter indicating possible permafrost at this site are worth further reflection and discussion.

Although you adapted your previous statements and analyses regarding the comparisons of weather data measured at Menara and on Toubkal, you did not attempt any of the other approaches suggested by Reviewer 2, such as land surface temperature data analysis, which would provide an interesting comparison with your point data and should at least be mentioned in the outlook.

As mentioned above, an important point I suggest you include and discuss is the potential distribution of avalanche snow and of rock fall debris (see Figure 11 for example), based on slope angles. Buried avalanche deposits may be the main / only type of permafrost remaining in this region

– independent of elevation but dependent on slope angle and snow / rock debris availability. The ice may be very old. Current avalanche deposits give clues to where snow may have been buried in the past. In the Swiss Alps we have for example observed excess ice at the base of slopes at low elevations (and hence not included on the Swiss permafrost distribution map), and buried under a thick layer of talus (see e.g. www.permos.ch – the Flüelapass site is one of these). Lambiel & Pieracci (2008) and (Scapozza et al., 2011) also provide such examples. You observed creeping features, lobate deposits and ridges/furrows at T3, which is at the base of a steep slope. These indicate that there is/was ice buried here – and if it is still present, it is a much more likely explanation for the low GSTs in winter than intra-talus ventilation. You should carefully consider what effects buried ground ice would have on the ground thermal regime. The presence of buried ice would also be of interest regarding future water resources in the High Atlas – an important aspect for further research in this arid area.

Some open questions: Did you notice any cold air fluxes at the base of the slope during very hot days in summer (pointing towards ventilation)? Are there any springs emanating from the lobes at T3 or elsewhere? Were spring temperatures measured? Have the local guides noticed a change in the occurrence of mass wasting?

**Further details:**

Title: I suggest you add '…the **possible** presence of permafrost…'

p. 3, line 17: is the atmosphere-soil interaction the major controlling factor on the ground thermal regime here? It may be one of the controlling factors – but has less relevance if you consider the presence of buried ice.

Study area: mention present/past snow avalanche activity

Methods: how were the temperature loggers calibrated? Ice-water mixture? Active layer zero-curtain?

Methods: remove the last sentence (iButtons would for example have been a cheap and practical solution – and why would there be a higher probability of disappearance at high altitude sites??)

3.2, p. 6, line 5: define the difference between 'snow' and 'significant snow' – and how is it determined?

Figure 4: where is the extrapolated data for D. Toubkal? (Figure 12?)

Figure 9: please label the axes in English (months)

What are your thoughts on the usefulness and quality of the remote sensing snow cover data?

Conclusions: please make these more concise and add a separate section: Outlook.

The study area is of great interest and in future the use of more measurement devices and different investigation techniques may well allow to gain a better insight on the presence and distribution of

permafrost here. However, at present there is very little data available and so any existing clues from other sources of information must be used to explain this data and to provide convincing arguments for the paper – and also to justify future research. The points mentioned above therefore need to be taken into account and addressed before your paper can be reconsidered for publication in the special issue of The Cryosphere.

**References**

Kenner, R., Magnusson, J., 2017. Estimating the Effect of Different Influencing Factors on Rock Glacier Development in Two Regions in the Swiss Alps. Permafrost and Periglacial Processes, 28(1), 195-208. DOI: 10.1002/ppp.1910

Scapozza, C., Lambiel, C., Baron, L., Marescot, L., Reynard, E., 2011. Internal structure and permafrost distribution in two alpine periglacial talus slopes, Valais, Swiss Alps. Geomorphology, 132(3–4), 208-221. DOI: http://dx.doi.org/10.1016/j.geomorph.2011.05.010

---

## Author Response (AR2)

Lisbon, 30 April 2017

Dear Dr Marcia Phillips,

Thank you very much for the careful review, comments and insights on our manuscript on GST and permafrost in the High Atlas. We have tried to answer your requests in this new version of the manuscript and we think that with the new informations, the text is more solid and we hope that it can be accepted for publication in the special issue of The Cryosphere.

Please find below a full copy of your comments and questions, which we have intercalated with our replies.

We have submitted both the clean version of the new manuscript, as well as a version with « track changes » as a supplement file.

Best wishes,

Gonçalo Vieira, Carla Mora and Ali Faleh

**Comments by the Editor and replies**

Prof. E. Serrano (Reviewer 1) commented that the analysis of the landforms was poor - and although you describe some of them, it is impossible for the reader to locate these, as there is no overview of them. A simple geomorphological map, an aerial photograph or even an excerpt from Google Earth with labelled landforms would be useful. This is particularly important in this setting, because the distribution of possible permafrost does not appear to be linked to an elevational gradient – but rather to morphological terrain characteristics. The process of ice genesis at the base of steep slopes due to mass wasting (avalanches, rockfall) needs to be discussed further and will be mentioned again below.

> *Reply : We have prepared a geomorphological sketch map, now included as figure 2, allowing to better discuss on the geomorphological features and their relation with measured data. Using the new map, we have also improved the description of the study area in section 2, including more information on mass wasting processes. The map also identifies the main scarps and differentiates between talus slopes, debris-mantled surfaces, typical of the higher reaches of the study area and the areas with deformation features.*

Dr. B. Staub (Reviewer 2) suggested carrying out a GIS analysis to investigate potential solar radiation, slope angles and terrain curvature. This would allow interesting comparisons with GIS analyses carried out in other mountain regions, e.g. (Kenner and Magnusson, 2017) and would help to determine whether the presence of permafrost is possible or not. From a purely morphological point of view, a simple GIS analysis would allow to determine locations where snow avalanches and rock fall can be deposited – thus pointing towards potential locations of buried ice at the foot of slopes (where air temperature regimes, solar radiation and GST have less influence on the distribution of permafrost than the presence of massive ice buried under a layer of rock debris).

*Reply : We have added a GIS analysis of potential radiation and slope angle as suggested and included it in the methodology, results and discussion. The results serve essentially as reference, since due to the scarce number of sites with GST data, we cannot realistically approach any modelling of relationships between measurements and GIS variables. The discussion on the possible presence of buried-ice has been added to the manuscript, but our observations do not provide clear support to this hypothesis. There are no clear avalanche tracks, nor gullies and the slope above T3 is not the pronest to snow avalanching, the same having been confirmed by our guide. There isn't also any evidence of thermokarst showing degradation of buried ice, nor any spring in June-July close to the arcuate features. The guide indicated that the snow accumulation present in the area results essentially from wind blown snow and we have advanced with the hypothesis of this being a form for ice to enter the talus.*

Reviewer 2 also suggested including freezing and thawing degree-days – this would be a further interesting way of comparing the GST regime at the 4 sites and of making first steps towards the development of High Atlas-specific GST indices – if they have any relation to permafrost distribution at all here. You state that you have these values and I suggest you include them. The extremely high summer GSTs at site T3 and the low ones in winter indicating possible permafrost at this site are worth further reflection and discussion.

*Reply : We have calculated the freezing and thawing degree-days and included them in Table 2. The results are presented in p.5, line 5-12. and also included the discussion, which is now substantially different.*

Although you adapted your previous statements and analyses regarding the comparisons of weather data measured at Menara and on Toubkal, you did not attempt any of the other approaches suggested by Reviewer 2, such as land surface temperature data analysis, which would provide an interesting comparison with your point data and should at least be mentioned in the outlook.

*Reply : We sincerely do not think that remote sensing LST will bring significant insight into assessing permafrost distribution, since LST are collected in the early morning, suffer from significant shadowing effects and are essentially an instantaneous snapshot of GST. Such GSTs vary enormously and very possibly will not correlate well with MAGST. Furthermore LST will only work during the snow free season and in that part of the year, GSTs are normally quite high and show no relation with MAGST. We therefore decided that there is no significant added-value in including this approach in the Outlook section.*

As mentioned above, an important point I suggest you include and discuss is the potential distribution of avalanche snow and of rock fall debris (see Figure 11 for example), based on slope angles. Buried avalanche deposits may be the main / only type of permafrost remaining in this region – independent of elevation but dependent on slope angle and snow / rock debris availability. The ice may be very old. Current avalanche deposits give clues to where snow may have been buried in the past. In the Swiss Alps we have for example observed excess ice at the base of slopes at low elevations (and hence not included on the Swiss permafrost distribution map), and buried under a thick layer of talus (see e.g. www.permos.ch – the Flüelapass site is one of these). Lambiel & Pieracci (2008) and (Scapozza et al., 2011) also provide such examples. You observed creeping features, lobate deposits and ridges/furrows at T3, which is at the base of a steep slope. These indicate that there is/was ice buried here – and if it is still

present, it is a much more likely explanation for the low GSTs in winter than intra-talus ventilation. You should carefully consider what effects buried ground ice would have on the ground thermal regime. The presence of buried ice would also be of interest regarding future water resources in the High Atlas – an important aspect for further research in this arid area.

> *Reply : We have further discussed the avalanching hypothesis (was already mentioned in the previous version) and accomodated the buried-ice genesis hypothesis relating it to rockfalls, in order to keep it as an open possibility. We have also improved the description of the arcuate features, which are quite incipient. However, with the current data, we cannot go much further in the discussion, than to present and discuss the various hypothesis, and this is what we did in several parts of the manuscript.*

**Some open questions:**

Did you notice any cold air fluxes at the base of the slope during very hot days in summer (pointing towards ventilation)?

> *Reply : No, we didn't. As we say in the text, the talus is not openwork, but matrix-supported.*

Are there any springs emanating from the lobes at T3 or elsewhere?

> *Reply : No, the area was dry during the two visits and we only found water in a small creek in the valley floor away from T3. This water should have been essentially derived from snow melt.*

Were spring temperatures measured?

> *Reply : No. We did not find springs close to the talus.*

Have the local guides noticed a change in the occurrence of mass wasting?

> *Reply : We haven't collected information on this issue, but is a good point, which we will adress in person with the guides in the future, since by email it is difficult to obtain accurate information.*

**Further questions**

Title: I suggest you add '…the possible presence of permafrost…'

> *Reply : Done.*

p. 3, line 17: is the atmosphere-soil interaction the major controlling factor on the ground thermal regime here? It may be one of the controlling factors – but has less relevance if you consider the presence of buried ice.

> *Reply : We have modified the sentence to « The detailed analysis of the GST provides a good insight on the atmosphere-soil interaction, as one of the major controlling factors on the ground thermal regime." This phrasing opens up the possibility for other factors such as buried ice, that, although possible, is not supported by surficial evidence.*

Study area: mention present/past snow avalanche activity

*Reply : information about avalanches is absent for the Ikhibi South valley, although we have found references to avalanching in the northwest facing slopes on the other side of the ridge. However, there the slope is steeper and there are avalanche corridors. After consulting with our guide, he mentioned that avalanches do not occur at the T3 monitoring site and that snow accumulation is mainly due to sheltering and wind drift. We added the following sentence in p. 4, line 30 : « Oral information from local guides confirms that during the cold season snow accumulates due to wind redistribution from the col and that avalanches do not occur at the site."*

Methods: how were the temperature loggers calibrated? Ice-water mixture? Active layer zerocurtain?

*Reply : The active layer zero-curtain was used to calibrate the loggers, with actual differences found to be below the sensor error (from 0.02 to 0.08 ºC). We have added a sentence in p5, line 15 : «The data was corrected using the freezing zero-curtain, having resulted in correction from -0.02 to -0.08 ºC."*

Methods: remove the last sentence (iButtons would for example have been a cheap and practical solution – and why would there be a higher probability of disappearance at high altitude sites??)

*Reply : We have deleted the sentence. As a note, we have decided to use Hobo TidBit, 5x more expensive than ibuttons, due to their better accuracy, resolution and due to being more robust. The mention to the disappearence was meant to air temperature loggers, since one i-button installed at the top of Toubkal was stollen and only those installed close to shops or huts made it through the year.*

3.2, p. 6, line 5: define the difference between 'snow' and 'significant snow' – and how is it

determined?

*Reply: we have added the following sentence: "The later showed a homogeneous spectral signal of the snow surface across the areas surrounding the sensor sites, while the "snow" class showed still some spectral mixture, but clearly the presence of snow cover."*

Figure 4: where is the extrapolated data for D. Toubkal? (Figure 12?)

*Reply : Given the comments of the referees, we decided to remove the extrapolated data and to present only the measured data. However, we forgot to delete the mention to that in the caption. It is now fixed.*

Figure 9: please label the axes in English (months)

*Reply : OK. Done.*

What are your thoughts on the usefulness and quality of the remote sensing snow cover data?

*Reply : The type of remote sensing imagery we have used is of limited application, but here it was meant to provide a confirmation of the periods with snow cover at the different sites. Since it is an area with absent data on snow cover, we think this was an adequate way to approach the question. It also provides significant information on the overall patterns of snow covers and melting, with important information emphasising on the fast melt of snow in the ridges and south facing slopes, contrary to shady slopes and high valley floors.*

Conclusions: please make these more concise and add a separate section: Outlook.

*Reply : Done. We have shortened the conclusions and added Outlook as section 7.*

[revised manuscript text omitted]

---

## Author Response (AR3)

Lisbon, 3 June 2017

5 Dear Dr Marcia Phillips,

Thank you very much for the very detailed review on contents and text improvement suggestions on the manuscript « New observations indicate the possible presence of permafrost in North Africa (Djebel Toubkal, High Atlas, Morocco) ». These are very sincerely appreciated.

10 We are now submitting the manuscript after including your proposed changes. You may find below the document after adding up your suggested modifications and with a track of the changes that we have made after your comments have been inserted (the track changes of included comments is not visible). Your questions and remarks are also commented below.

The main changes that we have made were the following :

- 15 Inclusion of almost all your suggestions for text changes (the few which were not included are explained in the track changes file),
  - Table 2 with data on freezing indices and synthesis on potential radiation for the monitoring sites was missing and is now included,
  - Incoming potential solar radiation in July had a small error, since the diffuse component was missing. This was corrected. The changes are small and do not have any implications for the interpretation of the data,
    - Added incoming potential solar radiation for December and a sentence in the text commenting it,
    - Kenner et al (2017) was included in the text in order to improve the interpretation of the results of site T3,
      - The conclusions were shortened and the main conclusions were indicated as bullet points,
      - Outlook was simplified,
      - Figure 1 was improved with a north arrow,
      - Figure 9 now includes also the potential solar radiation in December,
- 30 We hope you find that the changes are of added-value to the manuscript quality and look forward to

hearing from you.

Best wishes,

20

25

Gonçalo Vieira, Carla Mora and Ali Faleh

[revised manuscript text omitted]

**Comentado [mp2]: endemics ? Comentado [GV3R2]: ok**

Comentado [mp4]: Unclear. I suggest you remove this.

**Comentado [GV5R4]:** I would prefer maintaining. This would be linked to isotopic composition of ice and may provide interesting results in currently non glaciated mountains as another source of proxy data and if the paper is read by non-permafrost scientists, may open new lines for palecenvironmental research in the region. I have added « paleo..." to "environmental" no roder to clarify.

**Comentado [mp6]:** This is not really relevant here. I suggest you remove it.

**Comentado [GV7R6]:** OK, I have moved it further down since it provides a significant information from the area with known permafrost occurrence closer to the High Atlas.

Comentado [mp8]: Correct ? (a sort of intro to your findings...)
Comentado [GV9R8]: Right ! Thanks !

2,000 m a.s.l. The only landforms pointing towards a permafrost-related morphogenesis described in the literature are rock glaciers, as reported for the High Atlas by Dresch (1941), Wilche (1953) and Chardon and Riser (1981). Most of them are relict features and at least one case, the Arroumd rock deposit near Imlil, has recently been re-interpreted as a very large rock slide (Hughes et al., 2014). The only reference we found for active permafrost-related landforms is Chardon and Riser

- 5 (1981), who interpret a lobate feature in the Irhzer Ikbi South at 3,800 m a.s.l. as being an active rock glacier. However, there are no recent studies and the literature lacks direct observations and quantitative data indicating the presence of permafrost. To our knowledge, the only direct thermal observation of permafrost in the whole of Africa is from Mount Kilimanjaro, where permafrost has been reported at 5,785 m a.s.l. with ground temperatures of -0.03 °C at 3 m depth (Yoshikawa, 2013).
- 10 Given the climate change scenarios that the Mediterranean regions are facing, which are marked by warming and a decrease in precipitation (Giorgi and Lionello, 2007; Montanari, 2013; Simonneaux et al., 2015), permafrost should be close to disappearing in most Mediterranean mountains. The subsurface nature of permafrost and the presence of a thawed surface layer in the warmer season (the active layer) strongly limit its identification, characterization and mapping, especially in remote mountain areas (Gruber and Haeberli, 2009).
- 15 tOur study contributes towards solving the question of the presence of permafrost in North Africa and is an exploratory step towards an in-depth assessment aiming at the characterization and modelling of permafrost in the High Atlas. To carry out this initial assessment, we have installed a set of ground surface temperature (GST) and air temperature data loggers across an altitudinal gradient from 3,200 m to the summit of the Djebel Toubkal at 4,167 m in order toto characterize the ground temperature regime and heat exchange at the ground-atmosphere interface. The detailed analysis of the GSTs provides 20
- insight on the energy balance at the ground surface and its influence on the thermal regime of the underlying ground.

**2. Study Area**

The Djebel Toubkal is located in the Western High Atlas (31° 4' N, 7° 55' W) and is the highest mountain in North Africa at 4,167 m a.s.l. (Figure 1). The Atlas Mountains consist of a series of ranges and plateaus extending from southwest Morocco to northern Tunisia across more than 2,400 km (Mark and Osmaston, 2008). In Morocco, the Atlas Mountains include, from

- 25 north to south: the Middle Atlas (Djebel Bou Naceur, 3,340 m), the High Atlas (Djebel Toubkal, 4,167 m) and the Anti-Atlas (Djebel Sirwa, 3,304 m). The High and Middle Atlas are intracontinental fold-thrust belts located in the foreland of the Rif (Arboleya et al., 2004). The three major massifs in the High Atlas are, from west to east: the Djebel Toubkal Massif, the Irhil M'Goun Massif (4,071 m) and the Djebel Ayachi (3,751 m).

The climate in the High Atlas is influenced by the Atlantic Ocean to the west, the Mediterranean Sea to the nNorth and the Sahara Desert to the Scouth, resulting in a semi-arid to arid climate (Knippertz et al., 2003; Marchane et al., 2015). The rainy 30 season lasts from November to April and the dry season coincides with the summer, reflecting the Mediterranean style of the climate (N'da et al., 2016). Annual rainfall exceeds 600 mm above 700 m, and summer precipitation is mostly convective.

Comentado [mp10]: Please check that all the figure references appear in the correct order. Comentado [GV11R10]: ok

Comentado [mp12]: see comment below Comentado [GV13R12]: OK, all small caps Boudhar et al. (2014) report an average annual precipitation of 520 mm between 1989 and 2010 in Oukaimeden at 3,200 m elevation. Snow is present from November to April/May in the highest parts of the mountains, but with irregular regimes (Badri et al., 1994; Peyron, 1980) and is rarely continuous at mid-altitude, with snowfall and subsequent melt events sometimes happening within a few days. However, in the highest reaches, snow cover lasts for several weeks to months

- 5 (Boudhar et al., 2009). Snowmelt contributes to 15-50% of the stream flow in the Tensift catchment, thus playing a significant role for irrigation (Boudhar et al., 2009). The low atmospheric humidity and typically subfreezing temperatures above 3,000 m favour losses by sublimation, which can account up to 44% of snow ablation, while at lower altitudes melting prevails (Schulz and de Jong, 2004). The only perennial snow patch in North Africa is located below the north-facing cliffs of the Tazaghart plateau (3,980 m a.s.l.), close to the Toubkal. This feature is described in various recent papers and was
- 10 identified by Dresch (1941) together with other periglacial features (see Hughes, 2014). Its presence may be related to the high snow accumulation on the plateau above, together with the sheltering effect of the steep north-facing cliff face. The present study was conducted in the upper reaches of the Oued Inghyghaye valley, between the marabout of Sidi-Chamharouch and the summit of Djebel Toubkal (Figure 1). The lithology of the study area is characterized by Precambrian volcanics, such as Piroxene-bearing doleritic basalts and megaporphyric basalts of the Sidi Chamharouch formation (Zahour
- 15 et al., 2016) and andesites in the Djebel Toubkal (Cheggour, 2008; Rauh, 1952; Ros et al., 2000). The area has a typical alpine relief with sharp ridges rising above 3,500 m and long deep valleys. The upper catchments show evidence of Late Pleistocene glaciation with landforms such as roches mouttounées and moraines (Chardon and Riser, 1981; Hannah et al., 2016; Hughes et al., 2011; Hughes and Woodward, 2008; Mark and Osmaston, 2008; de Martonne, 1924). Extensive talus slopes and debris cones, together with widespread evidence of frost shattering mark the landscape above 3,000 m.
- 20 Our study area lies between the Neltner mountain hut of the Club Alpin Français de Casablanca and the Toubkal summit along the Irhzer Ikhibi-South valley, which is the main climbing route. The valley is about 1,400 m long and is a hanging tributary of the Oued Ihghyghayene valley, southeastwards of the Neltner hut above a rocky knoll with glacier polished outcrops at 3,350-3,400 m. Above the knoll, the valley floor is steep and is filled with accumulations of large boulders. Towards the east there are talus slopes, which are matrix-supported accumulations of decimetric to metric angular clasts
- 25 (figure 2). At ca. 3,800 m there are small lobate forms and incipient arcuate ramparts in the rock sediments, possibly indicating active periglacial dynamics, either by solifluction or permafrost creep. Chardon & Riser (1981) interpret this as an active rock glacier, but there is no steep front and no clearly defined debris body, so we do not support this interpretation. No springs have been observed near the talus slope. The valley has dissymmetric sides, with slope angles around 15-25° in the north-south-facing slope and 20 to 40° prevailing in the south north-facing slope. The valley headwalls are 30-40° steep, with
- 30 scarps and free rock faces occurring at 3,800-3,900 m in the aspects k yhorth and west facing slopes, except in the north facing sector of the slope, except in the north slope. Above the scarps, slope angles are 15-30° and the surfaces are mantled with angular clasts and boulders, with a similar situation in the north slope of the Irhzer Ikhibi-South. The southern ridge rises above 3,900 m, shading the valley floor in winter. Snow patches frequently remain in the valley until June, especially in the Toubkal slope, Local guides confirm that snow accumulates due to wind redistribution from the pass during the cold

Comentado [mp14]: correct ?

Comentado [GV15R14]: Yes, according to the Hughes and he is right.

| X                | Comentado [mp16]: north-facing ?                                              |
|------------------|-------------------------------------------------------------------------------|
|                  | Comentado [GV17R16]: no. I have clarified and changed to south-facing. |
| -                | Comentado [mp18]: south-facing ?                                              |
| 4                | Comentado [GV19R18]: north-facing.                                            |
| -                | Comentado [mp20]: please insert appropriate aspects                           |
| $\sum_{i=1}^{n}$ | Comentado [GV21R20]: OK, done.                                                |
|                  | Formatada: Não Realce                                                         |
| /                | Comentado [mp22]: unclear                                                     |
|                  | Comentado [GV23R22]: ok, deleted                                              |

season and that they have not observed avalanches at the site. Google Earth imagery allows identifying numerous debrismantled slopes and taluses with flow-like lineaments around the Toubkal, suggesting creep, and small rock glacier-like features can also be seen. There are solifluction lobes at the Irhzer Ikhibi Nord pass at 3,900 m, north of the Toubkal.

3. Methods

**5 3.1 Air and ground surface temperature monitoring**

Air temperature, relative humidity and ground surface temperature data loggers were installed in June 2015 from Sidi Chamharouch (2,370 m) to Djebel Toubkal (4,160 m a.s.l.) across an altitudinal transect, aiming to characterize the ground and air temperatures for 2015-16 at an hourly resolution. We used Hobo ProV2 loggers with an accuracy of  $\pm 0.2$  °C to measure air temperature and relative humidity, installed in radiation shields at ca. 2 m height. One was installed close to a

10 shop in Sidi Chamharouch (2,370 m) and the other near the Neltner hut (3,210 m). The positions of both sites were surveyed by local partners. A minilogger ibutton DS-1922L was installed at the summit of Djebel Toubkal (4,167 m), in the iron trig point at the summit. However, this logger disappeared, so no data is available.

To measure GST, single channel Hobo TidBit miniloggers with an accuracy of  $\pm 0.2$  °C were glued to the lower face of a 15x15x0.2 cm high diffusivity steel plate that maximizes contact with the soil particles, and buried at 2-3 cm depth (see

- 15 Ferreira et al. 2016). To check for temperature drift after retrieval, the loggers were tested at various temperatures (-20 °C to 39 °C) and showed average differences lower than 0.1 °C, which is well within sensor precision. The data was corrected using the freezing autumn zero-curtain as a reference temperature. Four such loggers were installed between the Neltner hut and the summit of Djebel Toubkal. The sites were selected to determine altitude control on GST All loggers were installed in stony soils with a silty-sandy matrix, in flat areas on otherwise gentle slopes. The Neltner (NLT) logger was installed
- 20 above the hut in a bouldery diamicton. Toubkal 3 (T3) was in a bouldery diamicton in a valley, where snow accumulates. Toubkal 2 (T2) was installed in a debris-mantled slope, a few metres below a ridge crest. Toubkal 1 (T1) was installed in the rock debris on the Djbel Toubkal summit plateau, about 100 m from the summit. Given the known high spatial variability of GST in complex mountain settings (see Gubler et al., 2011) the interpretation of the results must be conducted with care and this is why we tried to limit the differences in soil characteristics between sites. Details are provided in Table 1 and figure 3.
- 25 Temperatures were recorded hourly from 16 June 2015 to 16 July 2016 with the objective of having a whole year of data, with one cold season.

**3.2 Potential solar radiation modelling**

30

Slope angle and potential solar radiation were modelled using the algorithms provided by SAGA GIS 4.0, using a standard lumped atmospheric transmittance effect of 70% and the ASTER Global Digital Elevation Model version 2 from NASA, with a pixel size of 30 m. The total potential solar radiation (direct + diffuse) was calculated for the whole vear and also for

7

Comentado [mp24]: i.e. they may occur, but have not been observed

**Comentado [GV25R24]:** right. However, this area has many climbers almost everyday, even during winter and spring.

Comentado [mp26]: I moved this to the first reference to the hut Comentado [GV27R26]: excellent

Comentado [mp28]: you said this before already, not relevant here. Comentado [GV29R28]: ok

Comentado [mp30]: As you used GB spelling elsewhere
Comentado [GV31R30]: Ok. thanks

July and December, in order to roughly assess the influence of solar radiation energy input on the ground temperature maxima at the different sites.

**3.3 Remote sensing snow cover characterization**

- Although ground surface temperature regimes allow determining the presence of snow cover with a high degree of confidence, it was essential to demonstrate here that snow played a major role on GST and that it was present at some of the sites. To determine the presence/absence of snow cover we used 18 scenes from Landsat 8 OLI and 2 scenes from Sentinel 2-A, from 16/09/2015 to 14/06/2016. Landsat scenes were obtained at 16-day intervals, at 10 AM local time (USGS, 2016) and only one scene showed partial cloud cover. Sentinel-2 scenes at 10:30 AM complement the series and confirmed the Landsat results . Pixel size is 30 m for Landsat 8 and 10 m for Sentinel 2 (ESA, 2015). We used full resolution,
- 10 georeferenced visible colour composites provided by USGS at EarthExplorer. The images allowed to identify the general degree of snow cover at the monitoring sites. The interpretation had to be carried out with caution since the spatial resolution of the imagery is much larger lower than the microscale variability that may affect the monitoring sites. The snow conditions at each site were classified by visual inspection of the imagery as « no snow », « possible snow/snow margin », « snow » and « significant snow ». The latter showed a homogeneous spectral signal of the snow surface across the areas surrounding the sensor sites, while the "snow" class still showed some spectral mixture, but clearly indicated the presence of snow. The classification was done on-screen in QGIS, using an overlay of elevation contours and the coordinates of the monitoring sites for better accuracy. Differences between snow and cloud cover were easily identifiable and clouds were rare.

**3.4 Climate series and extrapolation**

- To put the period of June 2015 to July 2016 into a climate context, a long-term series of temperature and precipitation from a nearby meteorological station is needed. This allows comparing monthly records with the reference series in order to better frame the study period and discuss the results. However, the High Atlas has no long-term meteorological stations and the regional network is very sparse. The only long-term meteorological data available are from Marrakesh (Menara) in the plains north of the mountain range at 468 m a.s.l. and about 65 km from the study site, or from Ouarzazate, in the southern piedmont, at 1,153 m a.s.l, but in a very dry setting. The Middelt station, located 350 km to the east of the Toubkal Massif
- 25 at 1,515 m a.s.l. has very incomplete data. We used the more complete data series from Menara to compare the climate characteristics of the study period with a longer time series. The data was obtained from the Custom Monthly Summaries of the Global Historical Climate Network (NCDC) for 1977-2016. Before 1977 there are several gaps in the series.

**Comentado [mp32]:** How about showing December too ? (to help explain GST at T3)

**Comentado [GV33R32]:** We have added December. Table 2 was missing and is now included, which shows this data for each site.

Comentado [mp34]: lower ? (i.e. rougher) Comentado [GV35R34]: you're absolutely right.

[revised manuscript text omitted]

- 15 The ground may have voids at depth, which could help explain the low temperatures, but this has not yet been verified. The GST regimes measured at Neltner (3,200 m) clearly indicate that permafrost is absent. The snow pack only settled in mid-February and GST became very stable around 0 °C, which shows that there is no heat sink at depth, but rather a warmer unfrozen ground. Snow lasted about a week longer than at T3, possibly due to local effects, such as shading or thicker snow accumulation associated with snow drifting.
- 20 The upper sites on the ridge (T2 3,980 m) and on the summit of Toubkal (T3 4,160 m) show short periods of stable GST during snow fall episodes, but long lasting zero-curtains did not occur after the initial one in late October, which was synchronous at T1, T2 and T3 (figure 11). After that episode, GSTs were irregular, ranging between ~1 and 8 °C, thus indicating the absence of snow. The lack of zero-curtain effects and the frequent freeze-thaw cycles at T2 confirm the absence of snow and indicate a very dry soil. At both sites, the data suggest that the ground remains frozen below the surface
- 25 during the cold season, however the The GST regime analysis does not allow assessing on the presence of permafrost. The very high daily GST ranges during the warm season can be explained by the high insolation, together with the scarce moisture and rocky nature of the soil. We have no data to explain the very high GSTs at T3 with confidence, but they should be a function of local differences in soil thermophysical properties, together with the concave setting of the site, which receives more reflected and emitted radiation from the surrounding slopes and also due to a wind shelter effect. Both T2 and T1 are in convex terrain and are very wind exposed sites, which may explain lower summer maxima than at T3.

It is possible that permafrost is present at T3, since the valley shows landforms typical of mountain permafrost, such as lobate deposits and poorly developed transverse ridges and furrows in the lower part of the scree slope (figure 13). The high altitude and sheltered location favouring wind driven snow accumulation and longer lasting snow cover, together with the talus setting, also favours ice-burial either by snow deposition (dry snow blown into the debris voids) or by mass wasting

**Comentado [mp42]:** Again, see Kenner et al. 2017, Geomorphology, DOI: 10.1016/j.geomorph.2017.04.011 - which describes exactly this phenomenon.

Comentado [GV43R42]: Thanks. I have also used it above.

Comentado [mp44]: You say this in the outlook.
Comentado [GV45R44]: right

Comentado [mp46]: Ground temperatures can never be above 0°C under snow. Comentado [GV47R46]: Right. Comentado [mp48]: Which data suggests this ? I suggest you remove this. Comentado [GV49R48]: OK, I have removed. It has stayed as an heritage from a previous version, were we tried to compare with air temperatures estimated from Neltner, but it is better to keep it out. Comentado [mp50]: What about albedo ? are the rocks darker here?

Comentado [GV51R50]: No, they are not darker. I would not mention albedo without further data.

with rockfall covering snow deposits and conserving the ice. This may have occurred a long time ago -and would have led to the formation of intersticial ice facilitating the creep of the frozen talus, and forming the observed solifluction and creep related features. Amplified cooling over the snow surface due to the concave setting and cold air drainage, as observed in the Tatra mountains, may also be a process controlling ground cooling. The longer lasting snow also provides moisture that may

5 refreeze at depth when percolating into the debris.

The extrapolation of the long-term climate records from Menara to the summit of Djebel Toubkal using a lapse rate of -0.59 °C.100 m-1, a similar approach to Hannah et al. (2016) for paleoequilibrium line altitude estimation in the region allows for an insight into the climate sensitivity of the high reaches of the Toubkal Massif. The climate data should be interpreted with care, but it is worth noting that the extrapolation of the warming trend observed in Marrakesh, represents a gradual shift towards positive MAAT at the summit of Toubkal (Figure 14). This suggests that any possible permafrost in the High Atlas

10 towards positive MAAT at the summit of Toubkal (Figure 14). This suggests that any possible permafrost in the High Atlas is likely to disappear if the trend continues.

**6. Conclusions**

The measurement of GST from June 2015 to July 2016 at four sites across an altitudinal transect in the Toubkal Massif has provided first data from the high altitude periglacial domain of the High Atlas. The measurement period was marked by high air temperatures in December and January, with very low precipitation from November to January, causing a late onset of the winter snow cover. GSTs measured between 3,210 and 4,160 m a.s.l. showed two contrasting periods: a hot season from late May to late September, and a long cold season, from mid-October to mid-April. This pilot study allows for the following conclusions:

20

The hot season was characterized by positive air temperatures at all sites and by very high daily GST amplitudes,4 with maxima reaching up to 40.1 °C-at Toubkal 3. This regime was controlled by the high energy input from solar radiation and by the dryness of the soil.

----The cold season was marked by either subzero GSTs or by frequent freeze-thaw cycles, depending on snow conditions.

 Frequent freeze-thaw cycles were registered at Neltner in a valley floor at 3,210 m during the start of the cold season, until a heavy snow fall event in mid-February, inducing GST to stabilize around 0 °C.

The hHigh altitude sites (T2 3,964 and T1 4,160 m) located in windswept areas (T2 3,964 and T1 4,160 m) had subzero GSTs with frequent freeze-thaw cycles, indicating the lack of an insulating snow cover. These conditions prevailed during the whole cold season, with mean monthly GSTs below 0 °C from November to March.

30

25

The Toubkal-T33 minilogger located in Irhzer Ikhibi-South valley at 3,820 m a.s.l. registered a remarkable GST regime. After an early onset of a stable snow pack in mid-October, GST decreased regularly until mid-December

**Comentado [mp52]:** OK ? (as no avalanches have been observed recently and because buried ice can be conserved for thousands of years...)

Comentado [GV53R52]: Yes, this is possible to occur. Thanks for adding.

**Comentado [mp54]:** Don't introduce a new concept here – and it's unclear what you mean. Either rephrase or remove.

Comentado [GV55R54]: OK ; I have edited. I keep Hannah et al to support the use of the lapse rate

**Comentado [mp56]:** Your most important conclusion is missing: i.e. the fact that this pilot project has shown that altitude is probably not the main driving factor influencing permafrost distribution but topography (and likely mass wasting) are. Try shortening the conclusions and presenting your most important

findings as a list of bullet points.

**Comentado [mp57]: Correct ?**

**Formatada:** Parágrafo da Lista, Com marcas + Nível: 1 + Alinhado a: 0.63 cm + Avanço: 1.27 cm

and then stabilized around ca. -5.8 °C. These conditions were only interrupted in lasting until late-March-during snow melt

- The-GST regime at T3 at the site indicates the likely presence of permafrost, which is supported by the presence of poorly developed arcuate ridges and furrows in the area. This valley site has a long lasting snow cover due to wind accumulation and moderate potential incoming solar radiation in winter, -which favours lower ground temperatures. The data and observations are very scarce and do not allow for conclusions on the origin of the lower temperatures at the site, nor at assessing their spatial significance. The low winter GSTs may result from the joint effects of snow, cold air flow within the talus and over the snow surface, as well as from the possible presence of intersticial/buried ice resulting from rockfall events on snow.
- 10 Topography, possibly together with site-specific mass-wasting conditions, overcomes the driving effect of altitude on GST in the Toubkal Massif.

These results should be interpreted with care and although the preliminary data suggests the presence of permafrost, more observations are needed for a moren accurate assessment. The presence of permafrost at 3,800 m in the Djebel Toubkal massif, could become spatially significant for the High Atlas range with its high mountains and complex topography in both the Toubkal and M'Goun ranges-

15

**7. Outlook**

20

5

For the continuation of this pilot study, forthcoming research should target at: i. Electrical resistivity surveying to identify the presence and quantify the volume of ground ice, ii. Installing a larger number of GST loggers in different settings around T3, iii) Installing an air temperature logger at T3 to assess the possible shading effects of the topography on air temperature and GSTs, and iv) the drilling of a borehole to monitor- ground temperatures at depth and possibly obtain ice samples.

The presence of ice bearing permafrost may have impacts on the local hydrology, and would be of ecological significance due to the possible presence of extremophiles. Soil samples would possibly allow the analysis of permafrost ice as a paleoenvironmental archive and dating it would help to elucidate the history of mass wasting at the site. Given the warming trend in the region, further research is needed at an interdisciplinary level, since possible permafrost remnants could quickly

25 disappear and may be the last ones in North Africa. On the other hand, if ground ice occurs and it may be conserved for a long time despite climate warming, due to large latent heat of ice .-

**Acknowledgements**

Special thanks are due to Philip Hughes, who provided valuable advice on routes and logistics for the Toubkal massif and information on the geomorphology of the region. Warmest thanks to Sebastião Vieira, who participated in the field season of

30 2015 and helped with the setting of the loggers, for his fantastic company and patience. Mustapha Asquarray (Dar Assarou)

Comentado [mp58]: (For dating purposes), see below. Comentado [GV59R58]: thanks Comentado [mp60]: No need to repeat what was said in the introduction Comentado [GV61R60]: ok

Comentado [mp62]: Remark: But if you find out that there is ground ice AND that it is old, this would be very interesting too because 
[revised manuscript text omitted]

5

| A       | Altitude   | MAGST | FT Cycles     | FDD | TDD  | Potential Sola | ar Radiatio | on (KWh.m -2 ) |
|----------------|------------|-------|---------------|-----|------|----------------|-------------|---------------------------|
|                | (m a.s.l.) | (°C)  | ( nr ) |     |      | Annual         | July        | December                  |
| Neltner (NLT)  | 3210       | 6.0   | 62            | 110 | 2317 | 1703           | 230         | 49                        |
| Toubkal 3 (T3) | 3816       | 3.2   | 45            | 779 | 1936 | 1892           | 246         | 59                        |
| Toubkal 2 (T2) | 3964       | 3.3   | 143           | 325 | 1542 | 2631           | 264         | 126                       |
| Toubkal 1 (T1) | 4127       | 2.8   | 96            | 499 | 1536 | 2432           | 271         | 141                       |

Table 2 – Ground temperatures, frost indices and modelled potential solar radiation in the monitoring sites of the Toubkal Massif. 4–

Formatada: Legenda

---

## Author Response (AR4)

Lisbon, 9 June 2017

Dear Dr Marcia Phillips,

15 Thank you very much for the final revisions and comments, which we have implemented in the manuscript that we are now submitting.

In what concerns to your comment in p. 11, line 29, regarding the contradiction between the fact that the talus at the logger site is matrix-supported, with material filling the voids, and the possibility for snow to enter the voids, this is true for the exact location of the logger and for the present-day. However, in the past, the talus might have been open-work, as happens

20 in some parts of the valley. In order to clarify, we have changed the text to "…favours ice-burial either by snow deposition in the past (dry snow blown into the debris voids) or by mass wasting with rockfall covering…".

We have also corrected a typo in the name Chamharouch in figure 5, as well as passed **ºC** to ºC in figures 7, 10 and 14.

25 Best wishes,

Gonçalo Vieira, Carla Mora and Ali Faleh

[revised manuscript text omitted]